# On Volume Minimization in Conformal Regression

**Batiste Le Bars** [1]  **Pierre Humbert** [2]

## Abstract

We study the question of volume optimality in split conformal regression, a topic still poorly understood in comparison to coverage control. Using the fact that the calibration step can be seen as an empirical volume minimization problem, we first derive a finite-sample upper-bound on the excess volume loss of the interval returned by the classical split method. This important quantity measures the difference in length between the interval obtained with the split method and the shortest oracle prediction interval. Then, we introduce `EffOrt`, a methodology that modifies the learning step so that the base prediction function is selected in order to minimize the length of the returned intervals. In particular, our theoretical analysis of the excess volume loss of the prediction sets produced by `EffOrt` reveals the links between the learning and calibration steps, and notably the impact of the choice of the function class of the base predictor. We also introduce `Ad-EffOrt`, an extension of the previous method, which produces intervals whose size adapts to the value of the covariate. Finally, we evaluate the empirical performance and the robustness of our methodologies.

## 1. Introduction

Conformal Prediction (CP) (Vovk et al., 2005) has recently been considered as one of the state-of-art technique to construct distribution-free prediction sets satisfying probabilistic coverage guarantees. Formally, consider a random variable $(X, Y) \in \mathcal{X} \times \mathcal{Y}$ and some miscoverage level $\alpha \in [0, 1]$, CP techniques construct a set-valued function $C : \mathcal{X} \to 2^{\mathcal{Y}}$ such that:

$$\mathbb{P}(Y \in C(X)) \geq 1 - \alpha . \tag{1}$$

[1]Univ. Lille, Inria, CNRS, Centrale Lille, UMR 9189, CRIStAL, F-59000 Lille [2]Sorbonne Université et Univ. Paris Cité, CNRS, LPSM, F-75005 Paris, France. Correspondence to: Batiste Le Bars <batiste.le-bars@inria.fr>.

*Proceedings of the 42$^{nd}$ International Conference on Machine Learning*, Vancouver, Canada. PMLR 267, 2025. Copyright 2025 by the author(s).

This is particularly useful when the user prefers to be confident with the range of values that $Y$ can take, rather than having only a single predicted scalar value. In Section 2.1, we give a short reminder on CP, and on the most important techniques to construct $C$ satisfying Eq. (1). While these techniques are completely distribution-free, making them quite powerful in practice, they still suffer from an important limitation: how can we be sure that the trivial prediction set $C(x) = \mathcal{Y}$ is not returned? Indeed, this prediction set necessarily satisfy the condition (1). To prevent this, theoretical analyses of CP methods typically include an upper-bound on the probability of coverage $\mathbb{P}(Y \in C(X))$. Such upper-bound tends to $1 - \alpha$ as the number of sample used to build $C$ grows, which somehow reflects that the prediction set cannot be the full support of $Y$. However, this is still insufficient as one may take $C(x) = \mathcal{Y}$ with probability $1 - \alpha$ and $C(x) = \emptyset$ with probability $\alpha$, resulting in a coverage exactly equal to $1 - \alpha$, but with an expected size of $(1 - \alpha)|\mathcal{Y}|$. Here, $|\mathcal{Y}|$ denotes the size of $\mathcal{Y}$ and will typically be infinite in regression settings where $\mathcal{Y} = \mathbb{R}$. Such a set is too large and therefore highly uninformative. Hence, the CP literature suggests to also look at the size of the predicted sets to measure the performance of CP methods. The smaller is a prediction set, the more *efficient* it is considered. However, most works do this analysis empirically, while very few has been focusing on the statistical control of the size of CP sets.

In this paper, we therefore propose to study when $C(x)$ is in fact a solution of an optimization problem of the form:

$$\min_{C} \ \mathbb{E}[\mu(C(X))] \tag{2}$$
$$\text{s.t.} \quad \mathbb{P}(Y \in C(X)) \geq 1 - \alpha ,$$

where $\mu$ is a measure of the volume of the set $C(x)$, typically the Lebesgue measure in regression problems, or the counting measure in classification. This optimization problem ensures that among all prediction sets $C(x)$ that satisfy the coverage condition (1), the volume of the returned set is also of minimal size. Looking at problem (2) instead of (1) alone is therefore more meaningful as it encapsulates the two key aspects of CP: coverage and efficiency.

### 1.1. Main Contributions

- After describing the problem in Section 2.2, in Section 3 we restrict the prediction sets to be intervals of constant size, and show in Section 3.1 that the calibration step of

split CP solves an empirical version of problem (2). This allows us to derive a **finite-sample bound on the excess volume loss** of the returned prediction set, namely on the volume difference between the learned and the oracle prediction sets.

- We then argue that for the learning step to be *efficiency-oriented*, the prediction function should minimize the $(1-\alpha)$-quantile of the absolute error. This motivates EffOrt, a new split CP approach that finds an empirical minimizer of such quantile. In Theorem 3.7, an **excess volume bound shows the joint impact of the learning and the calibration step**, supporting the intuition that more data-points should be dedicated to the learning step.

- In Section 4, we increase the class of prediction sets to intervals with length adaptive to the covariates value, and present Ad-EffOrt, an extension of the previous method. Finally, in Section 5, a set of synthetic data experiments illustrates the empirical performance and the **robustness of our approaches** on asymmetric and heavy-tailed distributions.

## 2. Background

### 2.1. Preliminaries on Split Conformal Prediction

In this section, we give some important reminders on CP, focusing on the split approach at the core of this paper (Papadopoulos et al., 2002).

Let us assume that we have access to a data set $\mathcal{D} = \{(X_i, Y_i)\}_{1 \leq i \leq n}$, that we split into a learning set $\mathcal{D}^{lrn}$ and a non-overlapping calibration set $\mathcal{D}^{cal}$, containing respectively $n_\ell \geq 1$ and $n_c \geq 1$ data points such that $n_\ell + n_c = n$.

The first step of split CP, referred to as the *learning step*, consists in finding a *base* predictor $f \in \mathcal{F}$ using the learning data set $\mathcal{D}^{lrn}$. This predictor, denoted $\hat{f}$, is then used to define a nonconformity score function $s = s_{\hat{f}} : \mathcal{X} \times \mathcal{Y} \to \mathbb{R}$, such that for a pair $(x, y) \in \mathcal{X} \times \mathcal{Y}$, $s_{\hat{f}}(x, y)$ measures the level of non-conformity of the point $(x, y)$ with respect to the base predictor $\hat{f}$. In other word, it measures how far is the true value $y$ from the prediction $\hat{f}(x)$. Whether we are in the regression or classification setting, many possible base predictors and score functions exist in the literature (see e.g. (Angelopoulos & Bates, 2023)). In Example 1, we recall the most widely used base predictors and associated score functions for conformal regression.

In the second step of split CP, referred to as the *calibration step*, we construct the prediction set. To this end, we first calculate the values of $s_{\hat{f}}$ taken on the calibration set $\mathcal{D}^{cal}$, called the nonconformity scores $S_i := s_{\hat{f}}(X_i, Y_i)$, $i \in [\![n_c]\!]$. Then, we compute the $\lceil (n_c+1)(1-\alpha) \rceil$-th smallest nonconformity score $\hat{q}_{1-\alpha} := S_{(\lceil (n_c+1)(1-\alpha) \rceil)}$, where $S_{(1)} \leq \ldots \leq S_{(n_c)}$, and we return the set-valued function

$\widehat{C} : \mathcal{X} \to 2^{\mathcal{Y}}$ such that $\forall x \in \mathcal{X}$:

$$\widehat{C}(x) := \left\{ y \in \mathcal{Y} : s_{\hat{f}}(x, y) \leq \hat{q}_{1-\alpha} \right\}. \quad (3)$$

In the case where $\lceil (n_c + 1)(1 - \alpha) \rceil > n_c$, we fix $\hat{q}_{1-\alpha} = +\infty$, meaning that we take the trivial prediction set $\widehat{C}(x) = \mathcal{Y}$. Stated differently, $\hat{q}_{1-\alpha}$ corresponds to the $(1 - \alpha)$-quantile of the data set $\{S_i\}_{i=1}^{n_c} \cup \{+\infty\}$. Quite remarkably, if we only assume that the scores $S_1, \ldots, S_{n_c}$ and $s(X, Y)$ are *exchangeable*, the set (3) satisfies condition (1) (Papadopoulos et al., 2002). Moreover, if the scores are continuous random variables, it can be shown that $\mathbb{P}(Y \in \widehat{C}(X)) \leq 1 - \alpha + 1/(n_c + 1)$. Note that this type of guarantees are referred as *marginal* because the probabilities are taken with respect to the test point $(X, Y)$ and the calibration set $\mathcal{D}^{cal}$.

*Example* 1. *(Conformal regressors).*

1. In the standard *Split CP* (Papadopoulos et al., 2002) the base predictor is a function $\mu$ in $\mathcal{F}$, a class of regression function. Typically, $\hat{\mu} = \arg \min_{\mu \in \mathcal{F}} \sum_{i=1}^{n_\ell} (Y_i - \mu(X_i))^2$. Then, the score function is taken to be the absolute residual $s(x, y) = |y - \hat{\mu}(x)|$. This gives the interval $\widehat{C}(x) = [\hat{\mu}(x) - \hat{q}_{1-\alpha}, \hat{\mu}(x) + \hat{q}_{1-\alpha}]$.

2. In *Locally-Weighted Conformal Inference* (Papadopoulos et al., 2008), an additional base predictor is added in order to have interval sizes that adapt to the value of $X$. More precisely, we have $f = (\mu, \sigma)$, with $\mu \in \mathcal{F}_1$, $\sigma \in \mathcal{F}_2$. $\hat{\mu}$ is fitted as above, and $\hat{\sigma}$ fits the residuals given $X = x$, i.e. $\hat{\sigma} = \arg \min_{\sigma \in \mathcal{F}_2} \sum_{i=1}^{n_\ell} (R_i - \sigma(X_i))^2$ where $R_i = |Y_i - \hat{\mu}(X_i)|$. Taking the scoring function $s(x, y) = |y - \hat{\mu}(x)|/\hat{\sigma}(x)$, the resulting prediction interval is given by $\widehat{C}(x) = [\hat{\mu}(x) - \hat{\sigma}(x)\hat{q}_{1-\alpha}, \hat{\mu}(x) + \hat{\sigma}(x)\hat{q}_{1-\alpha}]$.

3. In *Conformalized Quantile Regression* (CQR) (Romano et al., 2019), we have $f = (Q_{\alpha/2}, Q_{1-\alpha/2})$ where $Q_{\alpha/2} \in \mathcal{F}_1$ (respectively $Q_{1-\alpha/2} \in \mathcal{F}_2$) is a quantile regressor of $Y$ given $X = x$, of order $\frac{\alpha}{2}$ (respectively $1 - \frac{\alpha}{2}$). For instance, we take $\widehat{Q}_{\alpha/2} = \arg \min_{Q \in \mathcal{F}_1} \sum_{i=1}^{n_\ell} \rho_{\alpha/2}(Y_i - Q(X_i))$, where $\rho_{\alpha/2}$ is the "pinball" loss (Koenker & Hallock, 2001). $\widehat{Q}_{1-\alpha/2}$ is defined analogously with $\rho_{1-\alpha/2}$. Then, we take $s(x, y) = \max\{\widehat{Q}_{\alpha/2}(x) - y, y - \widehat{Q}_{1-\alpha/2}(x)\}$, which gives $\widehat{C}(x) = [\widehat{Q}_{\alpha/2}(x) - \hat{q}_{1-\alpha}, \widehat{Q}_{1-\alpha/2}(x) + \hat{q}_{1-\alpha}]$.

As our task here is not to be exhaustive on the CP literature, we refer to Vovk et al. (2005), Angelopoulos & Bates (2023), and Fontana et al. (2023) for in-depth presentations of CP and to Manokhin (2022) for a curated list of papers.

### 2.2. Problem Statement

In this work, we focus on conformal regression problems with $\mathcal{Y} = \mathbb{R}$. Precisely, we study when and how split

CP outputs prediction sets approximating the solution of Problem (2). Since we consider regression tasks, let us first re-write the latter optimization problem by replacing $\mu$ with the Lebesgue measure $\lambda : \mathcal{B}(\mathbb{R}) \to [0, +\infty]$, $\mathcal{B}(\mathbb{R})$ being the Borel $\sigma$-algebra on $\mathbb{R}$:

$$\min_{C \in \mathcal{C}_{\text{Borel}}} \mathbb{E}[\lambda(C(X))] \text{ s.t. } \mathbb{P}(Y \in C(X)) \geq 1 - \alpha, \quad (4)$$

where $\mathcal{C}_{\text{Borel}} := \{\text{Measurable functions } C : \mathcal{X} \to \mathcal{B}(\mathbb{R})\}$. In the following, we refer to $C^*$ as one minimizer of (4). Note that optimizing over all possible measurable functions in $\mathcal{C}_{\text{Borel}}$ can be difficult in practice but also sometimes useless. For instance, in the regression setting where $Y = f^*(X) + \mathcal{N}(0, \sigma^2)$, the distribution of $Y$ given $X = x$ is symmetric and has only one mode. The optimal $C^*(x)$ will thus necessarily be an interval centered at $f^*(x)$ (see the discussion below on closed-form expressions for $C^*$). In this simple case, we see that looking at the full set $\mathcal{C}_{\text{Borel}}$ is useless as one could only consider the set of functions $C(x)$ that outputs intervals.

In this work, we will restrict the space of research $\mathcal{C}_{\text{Borel}}$ in (4) to smaller sets of set-valued functions, namely those outputting intervals. Like in statistical learning theory, this restriction can be thought of as a source of *approximation error*. In other words, we would like the restricted set to be sufficiently complex so that it includes (one of) the function solving (4). If it is not the case, we face such an approximation error. Nevertheless, controlling this error is not the objective of this paper, as we are going to mostly focus on the *estimation error*, which comes from the fact that only an empirical version of (4) is going to be solved.

**On closed-form expressions for** (4). In some settings, we can derive oracle prediction sets solving (4). For instance, when there is no covariate $X$, we recover the Minimum Volume Set (MVS) estimation problem of Scott & Nowak (2005). In that case, if $Y$ admits a density $p_Y(y)$ with respect to $\lambda$, we can derive a closed-form expression for $C^*$ in terms of density level sets: $\exists t_\alpha \geq 0$ such that $C^* = \{y \in \mathbb{R} : p_Y(y) \geq t_\alpha\}$ as soon as $\lambda(\{y \in \mathbb{R} : p_Y(y) = t_\alpha\}) = 0$. Similarly, if we condition the expectation and the probability in (4) on $X = x$, and if $Y|X = x$ admits a conditional density $p_{Y|X}(y|x)$, we get $C^*(x) = \{y \in \mathbb{R} : p_{Y|X}(y|x) \geq t'_\alpha(x)\}$ for some $t'_\alpha(x) \geq 0$ (Polonik & Yao, 2000; Lei & Wasserman, 2014). This has led to a whole literature based on plug-in (conditional) density estimators, which is not the approach considered in this paper but which is worth mentioning.

## 2.3. Related Work

**Minimum Volume Sets and Density Level Sets estimation.** As mentioned above, problem (4) is strongly linked with the MVS estimation Problem (Scott & Nowak, 2005),

which is itself linked with the problems of support estimation (Schölkopf et al., 2001; Munoz & Moguerza, 2006) and density level sets estimation (Polonik & Yao, 2000). Despite the fact that these methods can all be used to construct prediction sets with a desired coverage level, their link with Conformal Prediction has received little attention in the past. Among the most well known works, we can mention those taking the idea of plug-in (conditional) density estimators mentioned above, on top of which they add a calibration step to obtain better coverage guarantees (Lei et al., 2013; Lei & Wasserman, 2014; Izbicki et al., 2022; Chernozhukov et al., 2021). However, their theoretical results regarding the size of the returned set are mostly asymptotic. More importantly, modern supervised learning and CP techniques are rarely based on a non-parametric estimation of the (conditional) density, which can be difficult in practice. They are mostly based on the learning of a (parametrized) prediction function that belongs to a set of hypotheses (see Example 1). Thus, the framework of Section 2.2, largely inspired by Scott & Nowak (2005), where we restrict the class of prediction sets to a smaller subset, seems more appropriate for the design and analysis of CP methods.

**Efficient Conformal Prediction.** Recently, the question of controlling the size of the learned prediction set and explicitly see this as a minimization objective has attracted a lot of attention. For instance in Yang & Kuchibhotla (2024) and Liang et al. (2024), the authors focus on efficiency-oriented model selection. Closer to our work, we can mention Stutz et al. (2022) and Kiyani et al. (2024), which consider an optimization problem similar to that of (4), with a focus on the optimization aspects and on relaxations of the problem. However, they do not provide statistical guarantees on the learned estimates. Finally, Bai et al. (2022) proposes a generalization of the split calibration step, where instead of a single quantile, multiple learnable parameters are optimized to minimize the size of the final prediction set.

Other recent studies (Dhillon et al., 2024; Zecchin et al., 2024) have analyzed the expected size of split CP sets, highlighting key factors such as the impact of the score function choice and the generalization properties of the base predictor. In Sharma et al. (2023), the authors derive a PAC-Bayesian upper bound on the expected CP set size, involving an empirical estimate of the size and a KL divergence term. They also propose an algorithm that modifies the calibration step of split CP to minimize their bound; however, they do not explicitly instantiate their bound for their proposed algorithm, leaving the practical implications unclear.

Our work differs from these approaches in several fundamental ways. First, our theoretical guarantees are PAC-based rather than PAC-Bayesian, eliminating the need for restrictive assumptions such as boundedness of the score

function or target variable $Y$. More crucially, our analysis introduces an upper-bound on the excess volume loss, explicitly quantifying how much larger the CP set is compared to an oracle reference. This perspective is absent in prior works, which focus only on bounding the expected CP set size without reference to an optimal baseline. Finally, it is worth mentioning Correia et al. (2024), which provides an information-theoretic perspective by linking CP set size to conditional entropy.

## 3. Restriction to Intervals with Constant Size

In this section, we restrict the space of research in Problem (4) to the class of prediction sets $\mathcal{C}_{\mathcal{F}}^{\text{const}} = \{C_{f,t}(\cdot) = [f(\cdot) - t, f(\cdot) + t]; f \in \mathcal{F}, t \geq 0\}$. This class is already quite interesting as it encapsulates the standard split CP regressor (see Example 1.1). Notice that for simplicity of exposition, and because it does not depend on $x$, in this section the expected size of $C_{f,t} \in \mathcal{C}_{\mathcal{F}}^{\text{const}}$ is simply denoted $\lambda(C_{f,t}) = 2t$.

### 3.1. Base Predictor $f \in \mathcal{F}$ is Given: Optimality of the Conformal Step

We first start in the setting where the base predictor $f$ is given, meaning that we do not consider the learning phase. Over $\mathcal{C}_{\mathcal{F}}^{\text{const}}$, the optimization problem (4) becomes:

$$\min_{t \geq 0} \quad 2t \quad \text{s.t.} \quad \mathbb{P}(|Y - f(X)| \leq t) \geq 1 - \alpha. \quad (5)$$

Denoting by $S = |Y - f(X)|$ the random variable of the absolute residual, the solution of the above optimization corresponds to the quantile of order $1 - \alpha$ of the random variable $S$. More formally, if we denote by $Q(\cdot; S): [0, 1] \to \mathbb{R}$ the quantile function of $S$, then the optimal value solving (5) is exactly $t^* = Q(1 - \alpha; S)$ and the associated optimal set is $C_{f,t^*}^{1-\alpha}(x) = [f(x) - t^*, f(x) + t^*]$.

Importantly, notice that the conformal step of the original split CP in fact solves an empirical version of the previous problem, but with a slightly increased coverage:

$$\min_{t \geq 0} \quad 2t \quad (6)$$
$$\text{s.t.} \quad \frac{1}{n_c} \sum_{i=1}^{n_c} \mathbf{1}\{|Y_i - f(X_i)| \leq t\} \geq \frac{(1-\alpha)(n_c + 1)}{n_c},$$

with solution $\hat{t} = S_{(\lceil (n_c+1)(1-\alpha) \rceil)}$ and associated set denoted $C_{f,\hat{t}}^{1-\alpha}(x) = [f(x) - \hat{t}, f(x) + \hat{t}]$. As mentioned above, $\hat{t}$ is the quantity computed during the calibration step of the split CP method (see Section 2.1). It corresponds to the empirical quantile function of $S$, defined by $\widehat{Q}(q; \{S_i\}_{i=1}^{n_c}) := \inf\{t : \frac{1}{n_c} \sum_{i=1}^{n_c} \mathbf{1}\{S_i \leq t\} \geq q\}$, evaluated at $(1-\alpha)(n_c + 1)/n_c$ instead of $1 - \alpha$ to be slightly more conservative. In other words, this means that, when

$f$ is given, the calibration step in split CP outputs a conservative empirical estimator of the oracle prediction interval solution of Problem (5).

From the theory of CP, we already know that $\mathbb{P}(Y \in C_{f,\hat{t}}^{1-\alpha}(X)) \geq 1 - \alpha$ (see e.g. Lei et al. (2018, Theorem 2.2)). It remains to study the excess volume loss of $C_{f,\hat{t}}^{1-\alpha}$ which is measured by the difference in length between $C_{f,t^*}^{1-\alpha}$ and $C_{f,\hat{t}}^{1-\alpha}$. To this aim, it is sufficient to study the difference between the empirical quantile $\hat{t} = \widehat{Q}((1-\alpha)\frac{n_c+1}{n_c}; \{S_i\}_{i=1}^{n_c})$ and the true quantile $t^* = Q(1 - \alpha; S)$, as done in the following proposition (proof in Appendix A.1).

**Proposition 3.1.** *Let $\hat{t} = \widehat{Q}((1-\alpha)\frac{n_c+1}{n_c}; \{S_i\}_{i=1}^{n_c})$ and $C_{f,\hat{t}}^{1-\alpha}$ the corresponding set. If the points in $\mathcal{D}^{cal}$ are i.i.d., and if $(n_c + 1)(1 - \alpha)$ is not an integer, then with probability greater than $1 - \delta$ we have:*

$$\lambda\left(C_{f,\hat{t}}^{1-\alpha}\right) \leq 2Q\left(1 - \alpha + \frac{1-\alpha}{n_c} + \sqrt{\frac{\log(2/\delta)}{2n_c}}; S\right). \quad (7)$$

Interestingly, the right-hand side of (7) also corresponds to the optimal length of a more conservative oracle, namely $\lambda\left(C_{f,t^*}^{1-\alpha+\beta_{n_c}}\right)$ with $\beta_{n_c} = \frac{1-\alpha}{n_c} + \sqrt{\frac{\log(2/\delta)}{2n_c}}$. This means that, with high probability, the empirical interval obtained with the conformal step is smaller than the smallest oracle interval with increased coverage $1 - \alpha + \beta_{n_c}$, and where $\beta_{n_c}$ is tending to 0 as $n_c$ grows.

Although interesting, the previous result does not really tell us how different is the size of the predicted interval compared with the oracle one. To obtain a finite-sample upper bound on this difference, we must consider some regularity assumption on the distribution of $S$, and more particularly on its quantile function.

**Assumption 3.2.** *(Regularity condition).* Let $S = |Y - f(X)|$. $\forall f \in \mathcal{F}, \forall \alpha \in (0, 1), \exists r, \gamma \in (0, 1]$ and $L > 0$ such that $Q(\cdot; S)$ is locally $(\gamma, L)$-Hölder continuous, i.e. $\forall q_1, q_2 \in [1 - \alpha - r, 1 - \alpha + r]$:

$$|Q(q_1; S) - Q(q_2; S)| \leq L|q_1 - q_2|^\gamma.$$

This type of regularity condition can notably be found in Lei et al. (2013); Yang & Kuchibhotla (2024), where it is used to obtain finite-sample bounds on the volume of the returned set. Given this assumption, we can derive the following corollary.

**Corollary 3.3.** *Let the conditions of Proposition 3.1 and Assumption 3.2 hold. If $n_c$ is large enough so that $\frac{1-\alpha}{n_c} +$*

$\sqrt{\frac{\log(2/\delta)}{2n_c}} \leq r$, *then with probability greater than* $1 - \delta$:

$$\lambda\left(C_{f,\hat{t}}^{1-\alpha}\right) \leq \lambda\left(C_{f,t^*}^{1-\alpha}\right) + 2L\left(\frac{1}{n_c} + \sqrt{\frac{\log(2/\delta)}{2n_c}}\right)^{\gamma} . \quad (8)$$

*Proof.* Direct application of Prop 3.1 with Assumption 3.2 and using the fact that $1 - \alpha \leq 1$. □

The previous corollary provides an excess volume upper-bound for $C_{f,\hat{t}}^{1-\alpha}$ compared to the oracle $C_{f,t^*}^{1-\alpha}$. This bound does not only confirm the asymptotic optimality of the conformal procedure when $f$ is given, but also provides a rate of convergence dominated by $\tilde{\mathcal{O}}(n_c^{-\gamma})$ when we get rid of constants and log factors. Although simple to be obtained, to our knowledge this type of bound has never been shown.

*Remark* 3.4. When the base predictor is given, all the previous study can be easily extended to the general CP nested set view of Gupta et al. (2022). For simplicity of exposition, this analysis is deferred to Appendix B.1

### 3.2. Base Predictor $f \in \mathcal{F}$ is *Not* Given: Sub-Optimality of the Least-Square Regressor

In the previous section we saw that, when $f$ is fixed, the calibration step of the split CP method corresponds to the minimization of the size of the interval, up to some statistical error. Now, we investigate how $f$ should be learned during the learning step to obtain a prediction interval of minimal size. Let us consider Problem (4) over $\mathcal{C}_{\mathcal{F}}^{\text{const}}$:

$$\min_{f \in \mathcal{F}, t \geq 0} \quad 2t \quad \text{s.t.} \quad \mathbb{P}(|Y - f(X)| \leq t) \geq 1 - \alpha . \quad (9)$$

By replacing $t$ with its optimal value as a function of $f$, i.e. $t^* = Q(1 - \alpha; |Y - f(X)|)$, we obtain what we call the $(1 - \alpha)$-QAE problem (Quantile Absolute Error):

$$\min_{f \in \mathcal{F}} Q(1 - \alpha; |Y - f(X)|) . \quad (10)$$

In words, this optimization problem tells us that $f$ should minimize the $(1 - \alpha)$-quantile of the distribution of $S = |Y - f(X)|$. This is quite natural, since this quantile is the one selected to build the prediction interval, and the smaller it is, the smaller the interval will be.

What this optimization problem also tells us is that taking $f$ as the minimizer of the Mean Squared Error (MSE) $\mathbb{E}[(Y - f(X))^2]$, denoted $\mu(x) = \mathbb{E}[Y|X = x]$, like it is suggested in classical split CP, is not generally optimal in terms of volume minimization, and one should rather take the minimizer of the $(1 - \alpha)$-QAE. Notice that, while in general the minimizer of the MSE does not match the one of the $(1 - \alpha)$-QAE, it does in some settings. For instance, in Lei et al. (2018, Section 3), the authors claim that if the residual distribution $Y - \mu(X)$ is independent of $X$ and

admits a symmetric density with one mode at 0, then taking $f = \mu$ is optimal, i.e. the minimizer of the MSE matches the minimizer of the $(1 - \alpha)$-QAE. However, this kind of assumptions can be quite strong in practice, reason why it is preferable to keep the minimization of the $(1 - \alpha)$-QAE as the main objective, since it is optimal on $\mathcal{C}_{\mathcal{F}}^{\text{const}}$ no matter the distribution of $(X, Y)$.

### 3.3. `EffOrt`: EFFiciency-ORienTed Split Conformal Regression

In this section, we propose a methodology to approach the oracle prediction set $C_{f^*,t^*}^{1-\alpha}(x) = [f^*(x) - t^*, f^*(x) + t^*]$, with $f^*$ the minimizer of the $(1 - \alpha)$-QAE (Problem (10)) and $t^* = Q(1 - \alpha; |Y - f^*(X)|)$. We place ourselves in the split conformal framework of Section 2.1, having access to a learning data set $\mathcal{D}^{lrn}$ used to learn $f$, and a calibration data set $\mathcal{D}^{cal}$. With a slight abuse of notation we will write $i \in \mathcal{D}^{lrn}$ or $\mathcal{D}^{cal}$ to indicate $(X_i, Y_i) \in \mathcal{D}^{lrn}$ or $\mathcal{D}^{cal}$.

The proposed methodology, referred to as `EffOrt`, consists in the following steps:

1. Learn $\hat{f} \in \arg\min_{f \in \mathcal{F}} \widehat{Q}(1 - \alpha; \{|Y_i - f(X_i)|\}_{i \in \mathcal{D}^{lrn}})$, i.e. minimize the empirical version of the $(1 - \alpha)$-QAE

2. Proceed to the calibration step, i.e. take $\hat{t} = \widehat{Q}\left((1 - \alpha)\frac{n_c+1}{n_c}; \{|Y_i - \hat{f}(X_i)|\}_{i \in \mathcal{D}^{cal}}\right)$

3. For any test point $X \in \mathcal{X}$, output the prediction interval $C_{\hat{f},\hat{t}}^{1-\alpha}(X) = [\hat{f}(X) - \hat{t}, \hat{f}(X) + \hat{t}]$

In `EffOrt`, the main difficulty is in the first step, where the empirical $(1 - \alpha)$-QAE must be minimized. Indeed, it does not have a closed-form solution, and if we want to use a gradient-based optimization algorithm, we must compute the gradient of the empirical $(1 - \alpha)$-QAE which not trivial, or might even not be clearly defined. In the following, we present a gradient-based optimization procedure inspired by Pena-Ordieres et al. (2020).

#### 3.3.1. OPTIMIZATION OF THE EMPIRICAL $(1 - \alpha)$-QAE

We assume that $f \in \mathcal{F}$ is parametrized by $\theta \in \Theta$, and for the sake of generality, we consider the problem:

$$\min_{\theta} \widehat{Q}(1 - \alpha; \{\ell(\theta; Z_i)\}_{i \in \mathcal{D}^{lrn}}) . \quad (11)$$

Here, $\ell : \Theta \times \mathcal{Z} \to \mathbb{R}$ is a loss function, taking as input a parameter $\theta$ and a data point $Z_i$. In the step 1 of `EffOrt`, $Z_i = (X_i, Y_i)$ and $\ell(\theta; Z_i) = |Y_i - f_\theta(X_i)|$.

To solve this problem, one natural idea is to use a gradient descent algorithm on $\widehat{Q}(1 - \alpha; \{\ell(\theta; Z_i)\}_{i \in \mathcal{D}^{lrn}})$. However, this function is not differentiable in $\theta$. We therefore follow the strategy of Pena-Ordieres et al. (2020) and consider a

smooth approximation of it. More precisely, we first approximate the empirical cumulative distribution function (cdf) $\widehat{F}(t,\theta) := \sum_{i \in \mathcal{D}^{lrn}} \mathbf{1}\{\ell(\theta; Z_i) \leq t\}$ by another function $\widetilde{F}_\varepsilon$ where the indicator is replaced by a smooth version of it:

$$\widetilde{F}_\varepsilon(t,\theta) = \sum_{i \in \mathcal{D}^{lrn}} \Gamma_\varepsilon(\ell(\theta; Z_i) - t) \, ,$$

where $\varepsilon > 0$ is a parameter of the approximation. One possible choice for $\Gamma_\varepsilon$ is given in Pena-Ordieres et al. (2020, Eq. (2.6)) and is detailed in Appendix C. Then, we define the smooth empirical quantile function by:

$$\widetilde{Q}_\varepsilon(q; (\ell(\theta; Z_i))_{i \in \mathcal{D}^{lrn}}) = \inf\{t \, : \, \widetilde{F}_\varepsilon(t,\theta) \geq q\} \, . \quad (12)$$

For a given $q$ and $\varepsilon > 0$, under mild assumptions on the loss function $\ell(\cdot)$, one can show that the gradient of Eq. (12) is well-defined and has a closed-form that can be used in a gradient descent algorithm. The full procedure is detailed in Appendix C.

### 3.4. Theoretical Analysis

In this last subsection, we theoretically analyze the performance of the prediction set output by EffOrt. We are interested in two types of guarantees: (i) a coverage guarantee and (ii) an excess volume loss guarantee like the one in Eq. (8). To this aim, we require the following assumption.

**Assumption 3.5.** There exists $\phi(\mathcal{F}, \delta, n) < +\infty$ such that with probability at least $1 - \delta$:

$$\sup_{\substack{t \geq 0 \\ f \in \mathcal{F}}} \left| \mathbb{P}(|Y - f(X)| \leq t) - \frac{1}{n} \sum_{i=1}^n \mathbf{1}\{|Y_i - f(X_i)| \leq t\} \right|$$
$$\leq \phi(\mathcal{F}, \delta, n) \, .$$

In this assumption, $\phi(\mathcal{F}, \delta, n)$ bounds the worst-case estimation error of $\mathbb{P}(|Y - f(X)| \leq t)$ using the empirical estimate $\frac{1}{n} \sum_{i=1}^n \mathbf{1}\{|Y_i - f(X_i)| \leq t\}$ over the whole function class $\mathcal{F}$ and for any value of $t$. Typically, $\phi(\mathcal{F}, \delta, n)$ will decrease with an increasing number of data points $n$ and increase as the *complexity* of $\mathcal{F}$ gets larger. In the following proposition, we explicitly derive a closed-form expression for $\phi(\mathcal{F}, \delta, n)$ when the function class $\mathcal{F}$ is finite.

**Proposition 3.6.** (Finite class $\mathcal{F}$). *If $|\mathcal{F}| < \infty$, then Assumption 3.5 is verified with $\phi(\mathcal{F}, \delta, n) = \sqrt{\frac{\log(2|\mathcal{F}|/\delta)}{2n}}$.*

Similarly to the "classical" statistical learning framework, where it is possible to obtain generalization bounds for infinite hypothesis classes, it is possible to derive other closed-forms for $\phi(\mathcal{F}, \delta, n)$ in the infinite case by involving complexity measures like VC dimensions or Rademacher complexities. This, along with the proof of Prop. 3.6, is discussed in Appendix B.2. We can now present our main theoretical result.

**Theorem 3.7.** *Let the conditions of Prop. 3.1 hold, with the points in $\mathcal{D}^{lrn}$ and $\mathcal{D}^{cal}$ being i.i.d., and $C_{\hat{f},\hat{t}}^{1-\alpha}(x)$ be the prediction interval output by EffOrt . If Assumption 3.2 and 3.5 are satisfied, the distribution of $Y$ is atomless, $n_c$ and $n_\ell$ are large enough so that $\frac{1-\alpha}{n_c} + \sqrt{\frac{\log(2/\delta)}{2n_c}} \leq r$ and $\phi(\mathcal{F}, \delta, n_\ell) \leq r$, then:*

*1. $\mathbb{P}(Y \in C_{\hat{f},\hat{t}}^{1-\alpha}(X)|\mathcal{D}^{lrn}) \geq 1 - \alpha$ a.s.*

*2. With probability greater that $1 - 2\delta$:*

$$\lambda\left(C_{\hat{f},\hat{t}}^{1-\alpha}\right) \leq \lambda\left(C_{f^*,t^*}^{1-\alpha}\right) \quad (13)$$
$$+ 2L\left(\frac{1}{n_c} + \sqrt{\frac{\log(2/\delta)}{2n_c}}\right)^\gamma + 4L\phi(\mathcal{F}, \delta, n_\ell)^\gamma$$

*Proof sketch - Details in Appendix A.2.* The first result is classical (Lei et al., 2018). Let us focus on the second one, proved with the following steps.

Step 1: In the first step of EffOrt, we are actually solving the empirical objective $\min_{f \in \mathcal{F}, t \geq 0} \{t$ s.t. $n_l^{-1} \sum_{i \in \mathcal{D}^{lrn}} \mathbf{1}\{|Y_i - f(X_i)| \leq t\} \geq 1 - \alpha\}$, with solutions denoted by $\hat{f}$ and $\hat{t}_{lrn}$. Using the theory of MVS estimation (Scott & Nowak, 2005), we can compare this solution to the oracle one. Indeed, by adapting the proof of Scott & Nowak (2005, Theorem 1), we can show that with probability greater than $1 - \delta$:

$$\mathbb{P}(Y \in C_{\hat{f},\hat{t}_{lrn}}^{1-\alpha}(X)|\mathcal{D}^{lrn}) \geq 1 - \alpha - \phi(\mathcal{F}, \delta, n_\ell) \quad (14)$$

and

$$\lambda\left(C_{\hat{f},\hat{t}_{lrn}}^{1-\alpha}\right) \leq \lambda\left(C_{f_{1-\alpha+\phi}^*,t_{1-\alpha+\phi}^*}^{1-\alpha+\phi}\right) \, , \quad (15)$$

where $\phi \equiv \phi(\mathcal{F}, \delta, n_\ell)$ and $C_{f_{1-\alpha+\phi}^*,t_{1-\alpha+\phi}^*}^{1-\alpha+\phi}$ denotes the optimal oracle interval with coverage increased by $\phi(\mathcal{F}, \delta, n_\ell)$. This tells us that after the learning step we already have some guarantees: (i) a high probability coverage guarantee, with a looser coverage decreased by $\phi(\mathcal{F}, \delta, n_\ell)$, (ii) an excess volume guarantee, ensuring that the volume of the learned interval is smaller than the optimal one with coverage increased by $\phi$. Interestingly, this also means that the conformal step allows to obtain an almost sure coverage guarantee, and to get rid of the statistical error due to $\phi(\mathcal{F}, \delta, n_\ell)$ in the coverage.

Step 2: From (15) we have $\hat{t}_{lrn} \leq t_{1-\alpha+\phi}^*$ and therefore $\hat{t} \leq t^* + \hat{t} - \hat{t}_{lrn} + t_{1-\alpha+\phi}^* - t^*$.

With (7) in Prop. 3.1, we have $\hat{t} \leq Q(1 - \alpha + \frac{1-\alpha}{n_c} + \sqrt{\frac{\log(2/\delta)}{2n_c}}; |Y - \hat{f}(X)|_{|\mathcal{D}^{lrn}})$. Moreover, from (14), $\hat{t}_{lrn} \geq Q(1 - \alpha - \phi(\mathcal{F}, \delta, n_\ell); |Y - \hat{f}(X)|_{|\mathcal{D}^{lrn}})$. Hence, thanks to Assumption 3.2, $\hat{t} - \hat{t}_{lrn} \leq L\left(\frac{1}{n_c} + \sqrt{\frac{\log(2/\delta)}{2n_c}}\right)^\gamma +$

$L\phi(\mathcal{F}, \delta, n_\ell)^\gamma$. It remains to bound $t^*_{1-\alpha+\phi} - t^*$. By definition, we have $t^*_{1-\alpha+\phi} = Q(1 - \alpha + \phi; |Y - f^*_{1-\alpha+\phi}(X)|)$, and $t^* = Q(1 - \alpha; |Y - f^*(X)|)$. Moreover, we notice that $t^*_{1-\alpha+\phi} \leq Q(1 - \alpha + \phi; |Y - f^*(X)|)$ since by definition $f^*_{1-\alpha+\phi}$ minimizes $Q(1-\alpha+\phi; |Y-f(X)|)$ over all $f \in \mathcal{F}$. Hence, $t^*_{1-\alpha+\phi} - t^* \leq L\phi(\mathcal{F}, \delta, n_\ell)^\gamma$, by Assumption 3.2. We conclude by combining everything. $\qquad\square$

To the best of our knowledge, Theorem 3.7 is one of the first to provide such a finite-sample upper bound on the excess-volume loss. It explicitly reveals the impact of the two split conformal steps of EffOrt. The two first error terms (involving $n_c$) match the bound of Corollary 3.3, and can be seen as the volume loss due to the calibration step. While the third term, with $\phi(\mathcal{F}, \delta, n_\ell)$, is the error due to the learning step. If we omit the dependence in $\delta$, $\phi(\mathcal{F}, \delta, n_\ell)$ will typically be in the form of $\sqrt{\frac{\mathrm{Compl}(\mathcal{F})}{n_\ell}}$, where $\mathrm{Compl}(\mathcal{F})$ measures the complexity of $\mathcal{F}$ (see Prop. 3.6 and Appendix B.2). In most settings, we have $\mathrm{Compl}(\mathcal{F}) \gg \log(1/\delta)$. Hence, the rate in Eq. (13) supports the important intuition that the learning step remains more important than the conformal step, at least in the sense that more data-points are needed to reach convergence. It is thus preferable to assign more points to the learning than the calibration.

## 4. Extension to Intervals with Adaptive Size

We now consider the case of prediction intervals whose size adapts to the value of $X$. Formally, we consider the class of prediction sets $\mathcal{C}^{\mathrm{adap}}_{\mathcal{F},\mathcal{S}} = \{C_{f,s}(x) = [f(x) - s(x), f(x) + s(x)] : f \in \mathcal{F}, s \in \mathcal{S}\}$, where $\mathcal{S}$ is a class of non-negative functions. Importantly, this class of prediction sets encapsulates the Locally-Weighted Conformal Inference and the CQR methods (see Examples 1.2 and 1.3).

### 4.1. Oracle Prediction Set and Conditioning over $X = x$

Following a similar reasoning as in Section 3, we could first consider $f$ fixed and derive a closed-form oracle expression for $s$ by solving Problem (4) with $\mathcal{C}_{\mathrm{Borel}}$ replaced by $\mathcal{C}^{\mathrm{adap}}_{\mathcal{F},\mathcal{S}}$. Unfortunately, contrary to the previous section, the solution of this problem does not have a direct expression.

For this reason, we propose to modify the problem so that $s$ admits an oracle closed-form expression which can be naturally estimated empirically. More precisely, we condition the optimization problem over the event $X = x$, where $x \in \mathcal{X}$. In that case, the problem becomes:

$$\min_{s \in \mathcal{S}} s(x) \text{ s.t. } \mathbb{P}(|Y - f(x)| \leq s(x)|X = x) \geq 1 - \alpha \quad (16)$$

This problem is more difficult than (10) as a *conditional* coverage constraint is now required, which is known to be harder to obtain in practice (Vovk, 2012; Lei & Wasserman, 2014). If $\mathcal{S}$ is sufficiently complex, Problem (16) has an

oracle close-form solution, which is given by the $(1 - \alpha)$-quantile of $|Y - f(X)|$ conditioned on $X = x$, denoted by $s^*(x) := Q(1 - \alpha; |Y - f(X)|_{|X=x})$. Interestingly, the function $s^*(x)$ is the quantile regression function of $|Y - f(X)|$ given $X = x$, and corresponds to the solution of $\min_{s \in \mathcal{S}} \mathbb{E}[\rho_{1-\alpha}(|Y - f(X)| - s(X))]$, where $\rho_{1-\alpha}$ is the pinball loss. Hence, a natural solution is to use an empirical plug-in estimator of $s^*$, i.e. minimizing an empirical version of the pinball risk, as suggested in the next section.

*Remark* 4.1. Another strategy could be to directly solve an empirical version of $\min_{f \in \mathcal{F}, s \in \mathcal{S}}\{\mathbb{E}[s(X)] \text{ s.t. } \mathbb{P}(|Y - f(X)| \leq s(X)) \geq 1 - \alpha\}$. This would allow deriving results similar to those of the previous section (see Appendix B.3), but solving it in practice can be challenging, notably because of the empirical coverage constraint. Notice that, although their objective is different from ours, Bai et al. (2022) face a similar optimization problem, where they propose a smooth and differentiable relaxation to solve it.

### 4.2. Ad-EffOrt

We now describe our second method, Ad-EffOrt, which extends EffOrt to prediction intervals with adaptive size. Like in EffOrt, we consider the split CP framework, having access to a learning dataset $\mathcal{D}^{lrn}$ used to learn the base predictors $f$ and $s$, and a calibration data set $\mathcal{D}^{cal}$. Ad-EffOrt consists in the following steps:

1. $\hat{f} \in \arg\min_{f \in \mathcal{F}} \widehat{Q}(1 - \alpha; \{|Y_i - f(X_i)|\}_{i \in \mathcal{D}^{lrn}})$

2. $\hat{s} \in \arg\min_{s \in \mathcal{S}} \frac{1}{n_\ell} \sum_{i \in \mathcal{D}^{lrn}} \rho_{1-\alpha}(|Y_i - \hat{f}(X_i)| - s(X_i))$

3. $\hat{t} = \widehat{Q}\Big((1 - \alpha)\frac{n_c+1}{n_c}; \{|Y_i - \hat{f}(X_i)| - \hat{s}(X_i)\}_{i \in \mathcal{D}^{cal}}\Big)$

4. For any test point $X \in \mathcal{X}$, output $C^{1-\alpha}_{\hat{f},\hat{s},\hat{t}}(X) = [\hat{f}(X) - \hat{s}(X) - \hat{t}, \hat{f}(X) + \hat{s}(X) + \hat{t}]$.

In the first two steps of Ad-EffOrt, we learn the model $f$ as in EffOrt and then fit the residuals using a quantile regression or order $1 - \alpha$. Note that, in those two steps, the same data are used to learn both the prediction model $f$ and the quantile regressor $s$, but we also might split the learning set in two. Then, in the third step (calibration), we take the quantile of $\{|Y_i - \hat{f}(X_i)| - \hat{s}(X_i)\}_{i \in \mathcal{D}^{cal}}$. This comes from the fact that the final prediction interval is in the form $[f(x) - s(x) - t, f(x) + s(x) + t]$, and, given the base predictors $(f, s)$, the smallest $t$ such that we satisfy the coverage is $Q((1 - \alpha); |Y - f(X)| - s(X))$. This claim is easily proved by following the analysis of Section 3.1.

The main limitation of Ad-EffOrt is the difficulty of providing a theoretical guarantee similar to that of Theorem 3.7. This is notably due to the fact that while $s$ is learned in order to obtain conditional guarantees, $f$ is learned as in EffOrt,

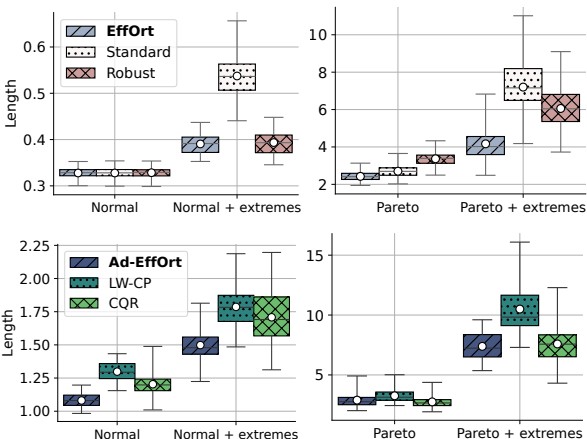

Figure 1. Boxplots of the 50 empirical expected lengths obtained by evaluating EffOrt in Section 5.1 (top) and Ad-EffOrt in Section 5.2 (bottom). The white circle corresponds to the mean.

i.e. in order to obtain marginal guarantees. When $f$ is fixed, one could actually derive guarantees on $\hat{s}$ and its ability to solve (16) by providing a setting under which the quantile regressor is consistent, making (16) asymptotically verified. Last but not least, it is worth mentioning that, thanks to the calibration step, the marginal coverage guarantee is verified.

## 5. Experiments

In this section we compare our methods, EffOrt and Ad-EffOrt, to the standard and locally adaptive versions of split CP on synthetic data. Due to lack of space, additional results on real data are deferred to Appendix D.2. Code to run all methods is available at https://github.com/pierreHmbt/AdEffOrt.

### 5.1. Evaluation of **EffOrt**

We first show the ability of EffOrt to return valid prediction sets of smaller size than those returned by standard split CP methods. More precisely, we consider asymmetric and heavy-tailed distribution, illustrating the robustness of our method to a wide range of realistic situations.

We consider a linear regression model $Y = X^T\theta + \mathcal{E}$ where $\theta \sim \mathcal{U}(0,1)^{\otimes 3}$ is fixed, $X \sim \mathcal{N}(0, I_3)$ with $I_3$ the identity matrix of size $3 \times 3$, and $\mathcal{E}$ follows 4 different distributions: A standard normal, a mixture distribution $0.95 \cdot \mathcal{N}(0,1) + 0.05 \cdot \mathcal{N}(2,1)$, a Pareto distribution with shape and scale parameters equal to 2 and 1, and another mixture equals to $0.95 \cdot \mathrm{Pareto}(2,1) + 0.05 \cdot \mathcal{N}(-20,1)$. In the two mixtures, the additional normal distributions allow simulating extreme values. For each scenario, we generate $n_{lrn} = n_{cal} = 1000$ pairs $(X_i, Y_i)$, as well as $n_{test} = 1000$ test points to compute the empirical marginal coverage and the average size of the returned set. We repeat this procedure 50 times.

During the learning step of EffOrt, we solve the $(1 - \alpha)$-QAE Problem (10) using the gradient descent strategy of Section 3.3.1. The smoothing parameter $\varepsilon$ is set to 0.1, $n_{iter} = 1000$, and the step-size sequence is $\{(1/t)^{0.6}\}_{t=1}^{n_{iter}}$. Furthermore, the space of research $\mathcal{F}$ is restricted to the space of linear functions (see Appendix D for additional results with Neural-Networks (NN)). For the split CP method, the regression function is either estimated using a linear regression or, in order to be fair in our comparisons, using a robust linear regression with Huber loss with parameter $\delta = 1.35$. For all methods, the score function is the absolute value of the residuals, i.e., $s(x, y) = |y - \widehat{f}(x)|$ and we set $\alpha = 0.1$.

**Results:** Figure 1 (top) displays the boxplots of the 50 test lengths obtained in the 4 scenarios (the coverage can be found in Appendix D and is, as expected, near 0.9). Overall, EffOrt produces more efficient marginally valid sets than those obtained with the split CP method, in all scenarios. Interestingly, when the noise follows a normal distribution (Figure 1 - top left panel), EffOrt and the split CP method with a standard or a robust linear regressor return similar sets. This was expected because with this type of distribution, the least-square regressor is supposed to be as good as the minimizer of the QAE. This is as opposed to the mixture of Gaussians, where extreme points brings asymmetry and makes the linear least square regression not suitable anymore. When the noise follows a Pareto distribution (Figure 1 top right panel), its heavy tail also makes the Split CP with robust regression output larger prediction sets. This could be explained by the fact that, in the learning step, the robust regression somehow gets rid of extreme points that should be kept, enforcing the calibration step to make a larger correction. In the last scenario, both baselines are outperformed by EffOrt.

### 5.2. Evaluation of **Ad-EffOrt**

We now compare Ad-EffOrt to the Locally Weighted CP (LW-CP) and CQR methods (see Example 1). We consider a simple heteroscedastic linear regression model $Y = X + \mathcal{E}(X)$ where $X \sim \mathcal{N}(0,1)$ and $\mathcal{E}(x)$ follows the 4 distributions of the previous section and with variance multiply by $x^2$. During the learning step of Ad-EffOrt, we solve the $(1 - \alpha)$-QAE Problem (10) using the gradient descent strategy of Section 3.3.1. The space of research $\mathcal{F}$ is restricted to the space of linear functions. We then learn $\hat{s}(\cdot)$ (second step of Ad-EffOrt) using a Random Forest (RF) quantile regressor. For the LW-CP method, the regression function is estimated using a linear regression and $\hat{\sigma}(\cdot)$ using a RF. Finally, for CQR, we also use a RF quantile regression. We set $\alpha = 0.1$. More details on the experimental setup are available in Appendix D.1.

**Results:** Figure 1 (bottom) displays the boxplots of the length for the 3 methods. The coverages can be found in

Appendix D and are near 0.9. Furthermore, an illustration of the returned sets is given in Figure 5 of Appendix D. We see that `Ad-EffOrt` returns valid marginal sets with length, on average, smaller or similar that the two other methods. Furthermore, the size of the boxplots are much smaller for our method than for the others. This means that `Ad-EffOrt` returns sets with more consistent sizes. Finally, we would like to point out that, although CQR gives similar results to our method in some situations (e.g., with the Pareto distribution), it has the drawback to not assess the uncertainty of a particular prediction model $\widehat{f}$.

## 6. Conclusion

This paper explicitly analyzes split conformal prediction through the lens of an MVS estimation problem and show that, in order to minimize the length of the prediction interval, the base predictor should minimize the $(1 - \alpha)$-QAE. This motivates two new methods, `EffOrt` and `Ad-EffOrt`, that are both empirically shown to be more robust than baselines over a significant spectrum of data-distributions. For `EffOrt`, a detailed theoretical analysis highlights how the complexity of the prediction function classes impacts the prediction interval's length. It also reveals that the calibration step allows to provide an almost sure coverage guarantee, at the cost of slightly increasing the excess volume loss, with a term dominated by the statistical error due to the learning step.

In the future, it would be interesting to explicitly control the error due to the approximation used in Section 3.3.1, and see its impact on Theorem 3.7. It would also be relevant to consider more complex classes of prediction sets, such as union of intervals, and to propose extensions of our framework to multivariate outputs. On a more general aspect, linking the volume of CP with more classical Uncertainty Quantification frameworks (Sale et al., 2023) is an interesting research direction.

## Acknowledgements

P. Humbert gratefully acknowledges the Emergence project MARS of Sorbonne Université.

## Impact Statement

This paper presents work whose goal is to advance the field of Machine Learning. There are many potential societal consequences of our work, none which we feel must be specifically highlighted here.

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

# Appendix

## A. Proofs of Main Results

In this section we give the proofs of the main results of the paper, starting with a reminder of the Dvoretzky–Kiefer–Wolfowitz (DKW) inequality used several times in the proofs.

**Lemma A.1.** (DKW inequality (Dvoretzky et al., 1956; Massart, 1990)) *Let $X_1, X_2, \ldots, X_n$ be real-valued independent and identically distributed random variables with cumulative distribution function $F(\cdot)$. Let $\hat{F}_n$ denote the associated empirical distribution function defined by $\hat{F}_n(x) = \sum_{i=1}^{n} \mathbf{1}\{X_i \leq x\}$. For all $\varepsilon > 0$:*

$$\mathbb{P}\Big( \sup_{x \in \mathbb{R}} |F(x) - \hat{F}_n(x)| > \varepsilon \Big) \leq 2e^{-2n\varepsilon^2} \ .$$

### A.1. Proof of Proposition 3.1

We have that $\lambda\Big(C_{f,\hat{t}}^{1-\alpha}\Big) = 2\hat{t}$. Let the events $E_1 := \Big\{\hat{t} > Q\Big(1 - \alpha + \frac{1-\alpha}{n_c} + \sqrt{\frac{\log(2/\delta)}{2n_c}}; S\Big)\Big\}$ and $E_{DKW} := \Big\{\sup_{t \geq 0} |F_S(t) - \hat{F}_S(t)| > \sqrt{\frac{\log(2/\delta)}{2n_c}}\Big\}$, where $F_S(t) = \mathbb{P}(S \leq t)$ and $\hat{F}_S(t) = \frac{1}{n_c}\sum_{i=1}^{n_c} \mathbf{1}\{S_i \leq t\}$. The main objective of the proof is to show that the event $E_1 \subset E_{DKW}$.

We first recall that $\hat{t} = \widehat{Q}((1-\alpha)\frac{n_c+1}{n_c}; \{S_i\}_{i=1}^{n_c})$, then $E_1$ is equivalent to:

$$\widehat{Q}\Big(1 - \alpha + \frac{1-\alpha}{n_c}; \{S_i\}_{i=1}^{n_c}\Big) > Q\Big(1 - \alpha + \frac{1-\alpha}{n_c} + \sqrt{\frac{\log(2/\delta)}{2n_c}}; S\Big) \ ,$$

which then implies that

$$1 - \alpha + \frac{1-\alpha}{n_c} > \hat{F}_S\Big(Q\Big(1 - \alpha + \frac{1-\alpha}{n_c} + \sqrt{\frac{\log(2/\delta)}{2n_c}}; S\Big)\Big) \ .$$

Where the last implication can be found for instance in the left-hand side of Eq. (34) in Howard & Ramdas (2022). It comes from the fact that $(1-\alpha)(n_c + 1)$ is not an integer and, in that case, $\widehat{Q}(\cdot; \{S_i\}_{i=1}^{n_c})$ acts as an inverse of $\hat{F}_S$.

Moreover, by definition of the quantile function and its relation with the cumulative distribution function (cdf), we have that

$$F_S\Big(Q\Big(1 - \alpha + \frac{1-\alpha}{n_c} + \sqrt{\frac{\log(2/\delta)}{2n_c}}; S\Big)\Big) \geq 1 - \alpha + \frac{1-\alpha}{n_c} + \sqrt{\frac{\log(2/\delta)}{2n_c}} \ .$$

Hence, $\Big|F_S\Big(Q\Big(1 - \alpha + \frac{1-\alpha}{n_c} + \sqrt{\frac{\log(2/\delta)}{2n_c}}; S\Big)\Big) - \hat{F}_S\Big(Q\Big(1 - \alpha + \frac{1-\alpha}{n_c} + \sqrt{\frac{\log(2/\delta)}{2n_c}}; S\Big)\Big)\Big| > \sqrt{\frac{\log(2/\delta)}{2n_c}}$, which implies $E_{DKW}$.

In the end, we have $\mathbb{P}(E_1) \leq \mathbb{P}(E_{DKW})$, and we conclude the proof by applying the DKW inequality from Lemma A.1 with $\varepsilon = \sqrt{\frac{\log(2/\delta)}{2n_c}}$.

### A.2. Proof of Theorem 3.7

We now detail the proof of our main result, which follows the sketch provided in the main text.

The first point of Theorem 3.7, on the almost sure coverage guarantee, is a classical result of the conformal prediction literature, see e.g. Lei et al. (2018, Theorem 2.2). Let us focus on the second result, which can be proved by following the two steps described hereafter.

**Step 1**: We first notice that in the first step of `EffOrt`, we are actually solving the empirical minimization objective:

$$\min_{f \in \mathcal{F}, t \geq 0} t$$
$$\text{s.t.} \quad \frac{1}{n_\ell} \sum_{i \in \mathcal{D}^{lrn}} \mathbf{1}\{|Y_i - f(X_i)| \leq t\} \geq 1 - \alpha \,,$$

with solutions denoted by $\hat{f}$ and $\hat{t}_{lrn}$.

Using the theory of MVS estimation (Scott & Nowak, 2005), we can compare this solution to the one of the oracle problem and show that with probability greater than $1 - \delta$:

$$\mathbb{P}(Y \in C^{1-\alpha}_{\hat{f}, \hat{t}_{lrn}}(X)|\mathcal{D}^{lrn}) \geq 1 - \alpha - \phi(\mathcal{F}, \delta, n_\ell) \tag{17}$$

and

$$\lambda\left(C^{1-\alpha}_{\hat{f}, \hat{t}_{lrn}}(X)\right) \leq \lambda\left(C^{1-\alpha+\phi}_{f^*_{1-\alpha+\phi}, t^*_{1-\alpha+\phi}}(X)\right) \,, \tag{18}$$

where $\phi \equiv \phi(\mathcal{F}, \delta, n_\ell)$ and $C^{1-\alpha+\phi}_{f^*_{1-\alpha+\phi}, t^*_{1-\alpha+\phi}}(X)$ denotes the optimal oracle interval with increased coverage $1 - \alpha + \phi(\mathcal{F}, \delta, n_\ell)$. In other word, $f^*_{1-\alpha+\phi}$ and $t^*_{1-\alpha+\phi}$ are the solutions of:

$$\min_{f \in \mathcal{F}, t \geq 0} t$$
$$\text{s.t.} \quad \mathbb{P}(|Y - f(X)| \leq t) \geq 1 - \alpha + \phi(\mathcal{F}, \delta, n_\ell) \,.$$

*Proof of* (17) *and* (18). Let:

- $\Theta_\mathbb{P} = \left\{\mathbb{P}(Y \in C^{1-\alpha}_{\hat{f}, \hat{t}_{lrn}}(X)|\mathcal{D}^{lrn}) < 1 - \alpha - \phi(\mathcal{F}, \delta, n_\ell)\right\}$

- $\Theta_\lambda = \left\{\lambda\left(C^{1-\alpha}_{\hat{f}, \hat{t}_{lrn}}(X)\right) > \lambda\left(C^{1-\alpha+\phi}_{f^*_{1-\alpha+\phi}, t^*_{1-\alpha+\phi}}(X)\right)\right\}$

- $\Theta_\phi = \left\{\sup_{t \geq 0, f \in \mathcal{F}}\left|\mathbb{P}(|Y - f(X)| \leq t) - \frac{1}{n_\ell}\sum_{i=1}^{n_\ell}\mathbf{1}\{|Y_i - f(X_i)| \leq t\}\right| > \phi(\mathcal{F}, \delta, n_\ell)\right\}$

The objective is to show that $\Theta_\mathbb{P} \cup \Theta_\lambda \subset \Theta_\phi$ since this would be mean that $\mathbb{P}(\Theta_\mathbb{P}^c \cap \Theta_\lambda^c) \geq \mathbb{P}(\Theta_\phi^c)$, where $\Theta^c$ is the complementary of $\Theta$. Then, applying Assumption 3.5 gives the desired result.

$\underline{\Theta_\mathbb{P} \subset \Theta_\phi}$: Consider $\Theta_\mathbb{P}$ is verified, i.e.

$$\mathbb{P}(Y \in C^{1-\alpha}_{\hat{f}, \hat{t}_{lrn}}(X)|\mathcal{D}^{lrn}) < 1 - \alpha - \phi(\mathcal{F}, \delta, n_\ell)$$

$$\implies \mathbb{P}(Y \in C^{1-\alpha}_{\hat{f}, \hat{t}_{lrn}}(X)|\mathcal{D}^{lrn}) - \frac{1}{n_\ell}\sum_{i=1}^{n_\ell}\mathbf{1}\{|Y_i - \hat{f}(X_i)| \leq \hat{t}_{lrn}\} < 1 - \alpha - \phi(\mathcal{F}, \delta, n_\ell) - \frac{1}{n_\ell}\sum_{i=1}^{n_\ell}\mathbf{1}\{|Y_i - \hat{f}(X_i)| \leq \hat{t}_{lrn}\}$$

$$\implies \mathbb{P}(Y \in C^{1-\alpha}_{\hat{f}, \hat{t}_{lrn}}(X)|\mathcal{D}^{lrn}) - \frac{1}{n_\ell}\sum_{i=1}^{n_\ell}\mathbf{1}\{|Y_i - \hat{f}(X_i)| \leq \hat{t}_{lrn}\} < -\phi(\mathcal{F}, \delta, n_\ell)$$

$$\implies \left|\mathbb{P}(Y \in C^{1-\alpha}_{\hat{f}, \hat{t}_{lrn}}(X)|\mathcal{D}^{lrn}) - \frac{1}{n_\ell}\sum_{i=1}^{n_\ell}\mathbf{1}\{|Y_i - \hat{f}(X_i)| \leq \hat{t}_{lrn}\}\right| > \phi(\mathcal{F}, \delta, n_\ell)$$

$$\implies \Theta_\phi$$

Where the second implication is obtained using the fact that by construction $\frac{1}{n_\ell}\sum_{i=1}^{n_\ell}\mathbf{1}\{|Y_i - \hat{f}(X_i)| \leq \hat{t}_{lrn}\} \geq 1 - \alpha$.

$\underline{\Theta_\lambda \subset \Theta_\phi}$:

Let us first show that $\Theta_\lambda$ implies that:

$$\frac{1}{n_\ell} \sum_{i=1}^{n_\ell} \mathbf{1}\{|Y_i - f_{1-\alpha+\phi}^*(X_i)| \le t_{1-\alpha+\phi}^*\} < 1 - \alpha \tag{19}$$

Indeed, if we had $\frac{1}{n_\ell} \sum_{i=1}^{n_\ell} \mathbf{1}\{|Y_i - f_{1-\alpha+\phi}^*(X_i)| \le t_{1-\alpha+\phi}^*\} \ge 1-\alpha$, then we would necessarily have $\hat{t}_{lrn} \le t_{1-\alpha+\phi}^*$, since $\hat{t}_{lrn}$ is minimal over the empirical coverage constraint, which would imply that $\lambda\left(C_{\hat{f},\hat{t}_{lrn}}^{1-\alpha}(X)\right) \le \lambda\left(C_{f_{1-\alpha+\phi}^*,t_{1-\alpha+\phi}^*}^{1-\alpha+\phi}(X)\right)$, i.e. that $\Theta_\lambda$ is not verified.

It remains to show that (19) implies $\Theta_\phi$. By (19), and using the fact that $\mathbb{P}\Big(|Y - f_{1-\alpha+\phi}^*(X)| \le t_{1-\alpha+\phi}^*\Big) \ge 1 - \alpha + \phi(\mathcal{F}, \delta, n_\ell)$:

$$\frac{1}{n_\ell} \sum_{i=1}^{n_\ell} \mathbf{1}\{|Y_i - f_{1-\alpha+\phi}^*(X_i)|\} - \mathbb{P}\Big(|Y - f_{1-\alpha+\phi}^*(X)| \le t_{1-\alpha+\phi}^*\Big) < -\phi(\mathcal{F}, \delta, n_\ell)$$

$$\implies \left|\frac{1}{n_\ell} \sum_{i=1}^{n_\ell} \mathbf{1}\{|Y_i - f_{1-\alpha+\phi}^*(X_i)|\} - \mathbb{P}\Big(|Y - f_{1-\alpha+\phi}^*(X)| \le t_{1-\alpha+\phi}^*\Big)\right| > \phi(\mathcal{F}, \delta, n_\ell)$$

$$\implies \Theta_\phi$$

This concludes the proof that $\Theta_\mathbb{P} \cup \Theta_\lambda \subset \Theta_\phi$ and therefore Eq. (17) and (18). $\qquad\square$

**Step 2**: From Eq. (7) in Prop. 3.1 we have that with probability greater than $1 - \delta$:

$$\hat{t} \le Q\left(1 - \alpha + \frac{1-\alpha}{n_c} + \sqrt{\frac{\log(2/\delta)}{2n_c}}; |Y - \hat{f}(X)|_{|\mathcal{D}^{lrn}}\right) \tag{20}$$

With an abuse of notation, we therefore have $\mathbb{P}(\{(20)\}) \ge 1 - \delta$ and $\mathbb{P}(\{(17)\} \cap \{(18)\}) \ge 1 - \delta$. Therefore, using the union bound over the complementary events, we get that $\mathbb{P}(\{(20)\} \cap \{(17)\} \cap \{(18)\}) \ge 1 - 2\delta$. In the following, we show that if (20), (17) and (18) are true, we have our final upper-bound, which will conclude the proof.

The size of the intervals being equal to 2 times their radius $t$, the objective here is to provide a high probability upper-bound on $\hat{t}$. Thanks to (18), we have that $\hat{t}_{lrn} \le t_{1-\alpha+\phi}^*$ and therefore:

$$\hat{t} = \hat{t} - \hat{t}_{lrn} + \hat{t}_{lrn} \le \hat{t} - \hat{t}_{lrn} + t_{1-\alpha+\phi}^* = t^* + \hat{t} - \hat{t}_{lrn} + t_{1-\alpha+\phi}^* - t^*$$

We first control $\hat{t} - \hat{t}_{lrn}$.

Applying the quantile function $Q(\cdot; |Y - \hat{f}(X)|_{|\mathcal{D}^{lrn}})$ on (17) gives $\hat{t}_{lrn} \ge Q(1 - \alpha - \phi(\mathcal{F}, \delta, n_\ell); |Y - \hat{f}(X)|_{|\mathcal{D}^{lrn}})$. Hence, thanks to (20) and to Assumption 3.2, we have:

$$\hat{t} - \hat{t}_{lrn} \le Q\Big(1 - \alpha + \frac{1-\alpha}{n_c} + \sqrt{\frac{\log(2/\delta)}{2n_c}}; |Y - \hat{f}(X)|_{|\mathcal{D}^{lrn}}\Big) - Q\Big(1 - \alpha - \phi(\mathcal{F}, \delta, n_\ell); |Y - \hat{f}(X)|_{|\mathcal{D}^{lrn}}\Big)$$

$$\le L\Big(\frac{1-\alpha}{n_c} + \sqrt{\frac{\log(2/\delta)}{2n_c}} + \phi(\mathcal{F}, \delta, n_\ell)\Big)^\gamma$$

$$\le L\Big(\frac{1}{n_c} + \sqrt{\frac{\log(2/\delta)}{2n_c}}\Big)^\gamma + L\phi(\mathcal{F}, \delta, n_\ell)^\gamma$$

It remains to bound $t_{1-\alpha+\phi}^* - t^*$. By definition, we have $t_{1-\alpha+\phi}^* = Q(1 - \alpha + \phi; |Y - f_{1-\alpha+\phi}^*(X)|)$, and $t^* = Q(1 - \alpha; |Y - f^*(X)|)$. Moreover, we notice that $Q(1 - \alpha + \phi; |Y - f_{1-\alpha+\phi}^*(X)|) \le Q(1 - \alpha + \phi; |Y - f^*(X)|)$ since by definition $f_{1-\alpha+\phi}^*$ minimizes $Q(1 - \alpha + \phi; |Y - f(X)|)$ over all $f \in \mathcal{F}$. In the end, by Assumption 3.2 we have:

$$t^*_{1-\alpha+\phi} - t^* \le Q(1-\alpha+\phi; |Y - f^*(X)|) - Q(1-\alpha; |Y - f^*(X)|) \le L\phi(\mathcal{F}, \delta, n_\ell)^\gamma .$$

We conclude the proof using the fact that $\lambda(C^{1-\alpha}_{\hat{f},\hat{t}}(X)) = 2\hat{t}$ and $\lambda(C^{1-\alpha}_{f^*,t^*}(X)) = 2t^*$. $\qquad\qquad\square$

## B. Additional Results

### B.1. The Nested Sets View

The split CP method described in Section 2.1 can also be described through the notion of *nested sets* (Gupta et al., 2022), which encapsulates many types of prediction sets, base predictors and scoring functions considered in the literature. As claimed in Remark 3.4, this framework will allow us to generalize the results of Section 3.1 to a wider class of prediction sets.

In the nested set view, we consider the class of prediction sets $\mathcal{C}^{\text{nested}}_{\mathcal{F},\mathcal{T}} = \{C_{f,t}(x) \text{ nested} ; f \in \mathcal{F}, t \in \mathcal{T} \subset \mathbb{R}\}$, where 'nested' means that for any fixed $f \in \mathcal{F}$ and $x \in \mathcal{X}$, $C_{f,t}(x) \subset C_{f,t'}(x)$ as soon as $t \le t'$. Here, we consider a fixed base predictor $f$, but as usual, $f$ is learned during the learning stage of the split method. In this setting, we can define the following general scoring function:

$$s_f(x, y) = \inf\{t \in \mathcal{T} : y \in C_{f,t}(x)\} .$$

Then, the procedure is the same as in Section 2.1: compute the nonconformity scores $S_i := s_f(X_i, Y_i)$, $i \in [\![n_c]\!]$ and find the $\lceil(n_c + 1)(1 - \alpha)\rceil$-th smallest one $\hat{q}_{1-\alpha} := S_{(\lceil(n_c+1)(1-\alpha)\rceil)}$. Finally, for any $x \in \mathcal{X}$, the prediction set is $C_{f,\hat{q}_{1-\alpha}}(x)$. As usual, the marginal guarantee is satisfied (Gupta et al., 2022, Prop. 1).

As mentioned above, an interesting aspect of this nested set framework is that it encapsulates many split CP approaches (Gupta et al., 2022, Table 1) such as the ones of Example 1, as shown in the following Example.

*Example 2. (Nested sets view of Example 1).*

1. The original *Split CP* (Papadopoulos et al., 2002) is recovered in the nested set framework by taking $f = \{\mu\}$, $C_{\mu,t}(x) = [\mu(x) - t; \mu(x) + t]$ and $\mathcal{T} = [0, \infty)$.

2. In *Locally-Weighted Conformal Inference* (Papadopoulos et al., 2008), $f = \{\mu, \sigma\}$, $C_{f,t}(c) = [\mu(x) - \sigma(x)t; \mu(x) + \sigma(x)t]$ and $\mathcal{T} = [0, \infty)$.

3. In *Conformalized Quantile Regression* (CQR) (Romano et al., 2019), we have $f = \{Q_\alpha, Q_{1-\alpha}\}$, $C_{f,t}(x) = [Q_\alpha(x) - t; Q_{1-\alpha}(x) + t]$ and $\mathcal{T} = \mathbb{R}$.

We can now extend our results from Section 3.1 to the nested framework, aiming at showing that, when $f$ is given, the conformal step indeed minimizes the size of the prediction set, up to an error that vanishes as $n_c$ grows.

To this aim, we need the following additional assumption on the way the size of the nested set grows with $t$.

**Assumption B.1.** *(Linear growth of the size.)* $\forall f \in \mathcal{F}, \exists a, b > 0$ such that $\mathbb{E}[\lambda(C_{f,t}(X))] = at + b$.

If we take the three previous examples, we have in 1) $a = 2$ and $b = 0$, 2) $a = 2\mathbb{E}[\sigma(X)]$ and $b = 0$, and 3) $a = 2$ and $b = \mathbb{E}[Q_{1-\alpha}(X) - Q_\alpha(X)]$.

Over $\mathcal{C}^{\text{nested}}_{\mathcal{F},\mathcal{T}}$ and under Assumption B.1, when $f$ is fixed the optimization problem (4) becomes:

$$\begin{aligned} &\min_{t \ge 0} \ at + b \\ &\text{s.t.} \quad \mathbb{P}(Y \in C_{f,t}(X)) \ge 1 - \alpha , \end{aligned}$$

which has the same solution as:

$$\begin{aligned} &\min_{t \ge 0} \ t \\ &\text{s.t.} \quad \mathbb{P}(s_f(X, Y) \le t) \ge 1 - \alpha , \end{aligned}$$

with solution $t^* = Q(1 - \alpha; s_f(X, Y))$. Similarly, the conformal step solves an empirical version of the previous problem:

$$\min_{t \geq 0} t$$

$$\text{s.t.} \quad \frac{1}{n_c} \sum_{i=1}^{n_c} \mathbf{1}\{s_f(X_i, Y_i) \leq t\} \geq (1 - \alpha)(n_c + 1)/n_c$$

with solution $\hat{t} = \widehat{Q}((1 - \alpha)(n_c + 1)/n_c; \{s_f(X_i, Y_i)\}_{i=1}^{n_c})$. As in Section 3.1, controlling the volume sub-optimality is equivalent to control the error of an empirical quantile estimate, and we can provide a very simple extension of Proposition 3.1 and Corollary 3.3.

**Proposition B.2.** *Let $\hat{t} = \widehat{Q}((1 - \alpha)\frac{n_c + 1}{n_c}; \{s_f(X_i, Y_i)\}_{i=1}^{n_c})$ and $C_{f,\hat{t}}(x)$ the corresponding (nested) prediction set. If Assumption B.1 holds, the points in $\mathcal{D}^{cal}$ are i.i.d., and $(n_c + 1)(1 - \alpha)$ is not an integer, then with probability greater than $1 - \delta$ we have:*

$$\mathbb{E}\left[\lambda\left(C_{f,\hat{t}}(X)\right) \middle| \mathcal{D}^{lrn}\right] \leq a \times Q\left(1 - \alpha + \frac{1 - \alpha}{n_c} + \sqrt{\frac{\log(1/\delta)}{2n_c}}; S\right) + b .$$

*Moreover, if Assumption 3.2 is true for $S = s_f(X, Y)$ and if $n_c$ is large enough so that $\frac{1 - \alpha}{n_c} + \sqrt{\frac{\log(1/\delta)}{2n_c}} \leq r$, then with probability greater than $1 - \delta$ we have:*

$$\mathbb{E}\left[\lambda\left(C_{f,\hat{t}}(X)\right) \middle| \mathcal{D}^{lrn}\right] \leq \mathbb{E}\left[\lambda\left(C_{f,t^*}(X)\right)\right] + aL\left(\frac{1 - \alpha}{n_c} + \sqrt{\frac{\log(1/\delta)}{2n_c}}\right)^\gamma .$$

*Proof.* The proof is essentially the same as the one of Proposition 3.1 and Corollary 3.3, and is therefore omitted. □

## B.2. Closed-Form Expressions for $\phi(\mathcal{F}, \delta, n)$

In Proposition 3.6 we give a closed form expression for $\phi(\mathcal{F}, \delta, n)$ in the case of finite function class $\mathcal{F}$. The proof is given hereafter.

*Proof of Proposition 3.6.* Let $\varepsilon > 0$,

$$\mathbb{P}\left(\sup_{t \geq 0, f \in \mathcal{F}} \left|\mathbb{P}(|Y - f(X)| \leq t) - \frac{1}{n} \sum_{i=1}^n \mathbf{1}\{|Y_i - f(X_i)| \leq t\}\right| > \varepsilon\right)$$

$$= \mathbb{P}\left(\bigcup_{f \in \mathcal{F}} \left\{\sup_{t \geq 0} \left|\mathbb{P}(|Y - f(X)| \leq t) - \frac{1}{n} \sum_{i=1}^n \mathbf{1}\{|Y_i - f(X_i)| \leq t\}\right| > \varepsilon\right\}\right)$$

$$\leq \sum_{f \in \mathcal{F}} \mathbb{P}\left(\sup_{t \geq 0} \left|\mathbb{P}(|Y - f(X)| \leq t) - \frac{1}{n} \sum_{i=1}^n \mathbf{1}\{|Y_i - f(X_i)| \leq t\}\right| > \varepsilon\right)$$

$$\leq 2|\mathcal{F}|e^{-2n\varepsilon^2} ,$$

where in the last inequality we use the DKW inequality, and the fact that $\mathcal{F}$ is finite. Finally, taking $\varepsilon = \sqrt{\frac{\log(2|\mathcal{F}|/\delta)}{2n}}$ concludes the proof. □

Other closed-form expressions can be obtained for infinite function classes using the classical notions of Rademacher complexity and VC dimension, as shown below.

Let $\widetilde{\mathcal{F}} = \{(x, y) \mapsto \mathbf{1}\{|y - f(x)| \leq t\} : f \in \mathcal{F}, t \geq 0\}$. The Rademacher complexity of the function class $\widetilde{\mathcal{F}}$ is the quantity

$$\mathcal{R}_n(\widetilde{\mathcal{F}}) = \mathbb{E}_{\mathcal{D}, \epsilon}\left[\sup_{f \in \mathcal{F}, t \geq 0} \frac{1}{n} \sum_{i=1}^n \epsilon_i \mathbf{1}\{|Y_i - f(X_i)| \leq t\}\right] ,$$

where $\epsilon_1, \ldots, \epsilon_n$ are Rademacher random variables. Then, a direct extension of the proof of Theorem 3.3 in Mohri (2018) gives the closed-form $\phi(\mathcal{F}, \delta, n) = 2\mathcal{R}_n(\widetilde{\mathcal{F}}) + \sqrt{\frac{\log(1/\delta)}{2n}}$.

It is also possible to bound the Rademacher complexity of $\widetilde{\mathcal{F}}$, first in terms of its associated Growth function (Massart's Lemma), and then in terms of its VC dimension, denoted $\text{VC}(\widetilde{\mathcal{F}})$ (Sauer's Lemma). Applying Corollary 3.8 and Corollary 3.18 in Mohri (2018) gives the closed-form $\phi(\mathcal{F}, \delta, n) = \sqrt{\frac{8\text{VC}(\widetilde{\mathcal{F}}) \log(en/\text{VC}(\widetilde{\mathcal{F}}))}{n}} + \sqrt{\frac{\log(1/\delta)}{2n}}$.

It should be noted that more informative close-forms could be obtained by specifying the function class of $\mathcal{F}$. For instance we could fix $\mathcal{F}$ to be the set of linear functions.

### B.3. Algorithm with Excess Volume Loss in the Adaptive Size Setting

In Section 4.1, if $\forall s \in \mathcal{S}$ and $t \geq 0$ we have $s + t \in \mathcal{S}$ (stability with addition of a scalar), then the oracle problem is equivalent to:

$$\min_{f \in \mathcal{F}, s \in \mathcal{S}, t \geq 0} \mathbb{E}[s(X)] + t$$
$$\text{s.t.} \quad \mathbb{P}(|Y - f(X)| - s(X) \leq t) \geq 1 - \alpha .$$

In practice, we propose to use Ad-EffOrt, however in order to obtain theoretical results similar to that of Theorem 3.7, another possibility would be to solve, during the learning step, an empirical version of the previous oracle problem, which is complicated to apply in practice:

$$\min_{f \in \mathcal{F}, s \in \mathcal{S}, t \geq 0} \frac{1}{n_\ell} \sum_{i \in \mathcal{D}^{lrn}} s(X_i) + t \tag{21}$$
$$\text{s.t.} \quad \frac{1}{n_\ell} \sum_{i \in \mathcal{D}^{lrn}} \mathbf{1}\{|Y_i - f(X_i)| - s(X_i) \leq t\} \geq 1 - \alpha - \phi(\mathcal{F}, \mathcal{S}, \delta, n_\ell) ,$$

where $\phi(\mathcal{F}, \mathcal{S}, \delta, n_\ell)$ is a penalty term relaxing the coverage constraint in order to obtain a smaller prediction set. This term corresponds to the statistical error of the empirical coverage, explicitly defined in the following assumption, which is necessary to derive a result similar to that of Theorem 3.7.

**Assumption B.3.** There exists two quantities $\phi(\mathcal{F}, \mathcal{S}, \delta, n) < +\infty$ and $\psi(\mathcal{S}, \delta, n) < +\infty$ such that:

$$\mathbb{P}\left(\sup_{f \in \mathcal{F}, s \in \mathcal{S}, t \geq 0} \left| \mathbb{P}(|Y - f(X)| - s(X) \leq t) - \frac{1}{n} \sum_{i=1}^{n} \mathbf{1}\{|Y_i - f(X_i)| - s(X_i) \leq t\} \right| \leq \phi(\mathcal{F}, \mathcal{S}, \delta, n) \right) \geq 1 - \delta$$

and

$$\mathbb{P}\left(\sup_{s \in \mathcal{S}} \left| \mathbb{E}[s(X)] - \frac{1}{n} \sum_{i=1}^{n} s(X_i) \right| \leq \psi(\mathcal{S}, \delta, n) \right) \geq 1 - \delta .$$

In words, this assumption generalizes Assumption 3.5 to the adaptive size setting, at least for the first equation. Since in this setting we also estimate the expectation of the size, we need the second equation to make sure that its worst-case estimation error is bounded w.h.p. Closed-form expressions for $\phi(\mathcal{F}, \mathcal{S}, \delta, n)$ and $\psi(\mathcal{S}, \delta, n)$ can be obtained similarly as in Appendix B.2.

Denote by $(\hat{f}, \hat{s}, \hat{t})$ the solutions of the empirical problem, $(f^*, s^*, t^*)$ the solutions of the oracle one, and $\forall (f, s, t)$, denote $C_{f,s,t}(x) = [f(x) - s(x) - t, f(x) + s(x) + t]$. Under Assumption B.3, we can derive the following lemma, which is an extension of the result obtained at the end of Step 1 in the proof of Theorem 3.7.

**Lemma B.4.** *Under Assumption B.3, we have with probability greater than $1 - 2\delta$:*

$$\mathbb{P}(Y \in C_{\hat{f}, \hat{s}, \hat{t}}^{1-\alpha}(X) | \mathcal{D}^{lrn}) \geq 1 - \alpha - 2\phi(\mathcal{F}, \mathcal{S}, \delta, n_\ell) \tag{22}$$

*and*

$$\mathbb{E}\left[ \lambda\left(C_{\hat{f}, \hat{s}, \hat{t}}^{1-\alpha}(X)\right) \Big| \mathcal{D}^{lrn} \right] \leq \mathbb{E}\left[ \lambda\left(C_{f^*, s^*, t^*}^{1-\alpha}(X)\right) \right] + 4\psi(\mathcal{S}, \delta, n_\ell) . \tag{23}$$

*Proof.* The proof closely follows the one of Step 1 in Appendix A.2. Let:

- $\Theta_{\mathbb{P}} = \left\{ \mathbb{P}(Y \in C^{1-\alpha}_{\hat{f},\hat{s},\hat{t}}(X)|\mathcal{D}^{lrn}) < 1 - \alpha - 2\phi(\mathcal{F},\mathcal{S},\delta,n_\ell) \right\}$

- $\Theta_{\lambda} = \left\{ \mathbb{E}\left[ \lambda\left( C^{1-\alpha}_{\hat{f},\hat{s},\hat{t}}(X) \right) \Big| \mathcal{D}^{lrn} \right] > \mathbb{E}\left[ \lambda\left( C^{1-\alpha+\phi}_{f^*,s^*,t^*}(X) \right) \right] + 4\psi(\mathcal{S},\delta,n_\ell) \right\}$

- $\Theta_{\phi} = \left\{ \sup_{f\in\mathcal{F},s\in\mathcal{S},t\geq 0} \left| \mathbb{P}\left(|Y-f(X)|-s(X)\leq t\right) - \frac{1}{n_\ell}\sum_{i\in\mathcal{D}^{lrn}}\mathbf{1}\{|Y_i-f(X_i)|-s(X_i)\leq t\} \right| > \phi(\mathcal{F},\mathcal{S},\delta,n_\ell) \right\}$

- $\Theta_{\psi} = \left\{ \sup_{s\in\mathcal{S}} \left| \mathbb{E}[s(X)] - \frac{1}{n_\ell}\sum_{i\in\mathcal{D}^{lrn}}s(X_i) \right| > \psi(\mathcal{S},\delta,n_\ell) \right\}$

The objective is to show that $(\Theta_{\mathbb{P}} \cup \Theta_{\lambda}) \subset (\Theta_{\phi} \cup \Theta_{\psi})$. Indeed, using the union bound and Assumption B.3, this would imply that $\mathbb{P}(\Theta_{\mathbb{P}} \cup \Theta_{\lambda}) \leq \mathbb{P}(\Theta_{\phi} \cup \Theta_{\psi}) \leq 2\delta$, concluding the proof.

$\underline{\Theta_{\mathbb{P}} \subset (\Theta_{\phi} \cup \Theta_{\psi})}$: Proved by showing that $\Theta_{\mathbb{P}} \subset \Theta_{\phi}$ using the same arguments as in the proof of the main result.

$\underline{\Theta_{\lambda} \subset (\Theta_{\phi} \cup \Theta_{\psi})}$:

Let the event $\Omega = \left\{ \frac{1}{n_\ell}\sum_{i\in\mathcal{D}^{lrn}}\mathbf{1}\{|Y_i-f^*(X_i)|-s^*(X_i)\leq t^*\} < 1-\alpha-\phi(\mathcal{F},\mathcal{S},\delta,n_\ell) \right\}$. We first show that $\Theta_{\lambda} \subset (\Omega \cup \Theta_{\psi})$, by proving that $(\Omega^c \cap \Theta_{\psi}^c) \subset \Theta_{\lambda}^c$. Indeed, under $(\Omega^c \cap \Theta_{\psi}^c)$ we have:

$$\frac{1}{n_\ell}\sum_{i\in\mathcal{D}^{lrn}}\mathbf{1}\{|Y_i-f^*(X_i)|-s^*(X_i)\leq t^*\} \geq 1-\alpha-\phi(\mathcal{F},\mathcal{S},\delta,n_\ell)$$

$$\implies \frac{1}{n_\ell}\sum_{i\in\mathcal{D}^{lrn}}\hat{s}(X_i) + \hat{t} \leq \frac{1}{n_\ell}\sum_{i\in\mathcal{D}^{lrn}}s^*_{1-\alpha+\phi}(X_i) + t^*$$

$$\implies \mathbb{E}[\hat{s}(X)|\mathcal{D}^{lrn}] + \hat{t} + \frac{1}{n_\ell}\sum_{i\in\mathcal{D}^{lrn}}\hat{s}(X_i) - \mathbb{E}[\hat{s}(X)|\mathcal{D}^{lrn}] \leq \mathbb{E}[s^*_{1-\alpha+\phi}(X)] + t^* + \frac{1}{n_\ell}\sum_{i\in\mathcal{D}^{lrn}}s^*_{1-\alpha+\phi}(X_i) - \mathbb{E}[s^*_{1-\alpha+\phi}(X)]$$

$$\implies \mathbb{E}[\hat{s}(X)|\mathcal{D}^{lrn}] + \hat{t} \leq \mathbb{E}[s^*_{1-\alpha+\phi}(X)] + t^* + 2\sup_{s\in\mathcal{S}}\left| \mathbb{E}[s(X)] - \frac{1}{n_\ell}\sum_{i\in\mathcal{D}^{lrn}}s(X_i) \right|$$

$$\implies \mathbb{E}[\hat{s}(X)|\mathcal{D}^{lrn}] + \hat{t} \leq \mathbb{E}[s^*_{1-\alpha+\phi}(X)] + t^* + 2\psi(\mathcal{S},\delta,n_\ell)$$
$$\implies 2\mathbb{E}[\hat{s}(X)|\mathcal{D}^{lrn}] + 2\hat{t} \leq 2\mathbb{E}[s^*_{1-\alpha+\phi}(X)] + 2t^* + 4\psi(\mathcal{S},\delta,n_\ell) \implies \Theta_{\lambda}^c .$$

It remains to prove that $\Omega \subset \Theta_{\phi}$. Under $\Omega$ and using the fact that $\mathbb{P}\left(|Y-f^*(X)|-s^*(X)\leq t^*\right) \geq 1-\alpha$:

$$\frac{1}{n_\ell}\sum_{i\in\mathcal{D}^{lrn}}\mathbf{1}\{|Y_i-f^*(X_i)|-s^*(X_i)\leq t^*\} - \mathbb{P}\left(|Y-f^*(X)|-s^*(X)\leq t^*\right) < -\phi(\mathcal{F},\mathcal{S},\delta,n_\ell)$$

$$\implies \left| \frac{1}{n_\ell}\sum_{i\in\mathcal{D}^{lrn}}\mathbf{1}\{|Y_i-f^*(X_i)|-s^*(X_i)\leq t^*\} - \mathbb{P}\left(|Y-f^*(X)|-s^*(X)\leq t^*\right) \right| > \phi(\mathcal{F},\mathcal{S},\delta,n_\ell)$$

$$\implies \Theta_{\phi} .$$

Hence $\Omega \subset \Theta_{\phi}$, i.e. $\Theta_{\lambda} \subset (\Omega \cup \Theta_{\psi}) \subset (\Theta_{\phi} \cup \Theta_{\psi})$, which concludes the proof. $\qquad\square$

Like after step 1 of `EffOrt`, with Lemma B.4 we have probabilistic guarantees on the coverage and on the expected size of the returned set. Using conformal prediction, we can now obtain an almost sure guarantee on the coverage, at the cost of slightly increasing the size of the set by $\hat{t}_c = \widehat{Q}\left( (1-\alpha)\frac{n_c+1}{n_c} ; \{|Y_i-\hat{f}(X_i)|-\hat{s}(X_i)\}_{i\in\mathcal{D}^{cal}} \right)$.

**Theorem B.5.** *Consider that Assumption 3.2 is satisfied for $S = |Y-f(X)|-s(X)$. Assume further that Assumption B.3 is verified, that the distribution of $Y$ is atomless, that $n_c$ and $n_\ell$ are large enough so that $\frac{1}{n_c+1} + \sqrt{\frac{\log(1/\delta)}{n_c+1}} \leq r$ and $\phi(\mathcal{F},\mathcal{S},\delta,n_\ell) \leq r$, then we have:*

1. $\mathbb{P}(Y \in C^{1-\alpha}_{\hat{f}, \hat{s}, \hat{t}_c}(X)|\mathcal{D}^{lrn}) \geq 1 - \alpha$ *a.s.*

2. *With probability greater that* $1 - 3\delta$:

$$\mathbb{E}\left[\lambda\left(C^{1-\alpha}_{\hat{f},\hat{s},\hat{t}_c}(X)\right)\Big|\mathcal{D}^{lrn}, \mathcal{D}^{cal}\right] \leq \mathbb{E}\left[\lambda\left(C^{1-\alpha}_{f^*,s^*,t^*}(X)\right)\right] + 4\psi(\mathcal{S}, \delta, n_\ell) + 2L\Big(\frac{1}{n_c + 1} + \sqrt{\frac{\log(1/\delta)}{n_c + 1}} + 2\phi(\mathcal{F}, \mathcal{S}, \delta, n_\ell)\Big)^\gamma. \tag{24}$$

*Proof.* Like in Theorem 3.7, the first point of Theorem B.5, on the almost sure coverage guarantee, is a classical result of the conformal prediction literature.

We start the proof of the second point by recalling that since the distribution of $Y$ is assumed atomless, we have with probability greater than $1 - \delta$:

$$\mathbb{P}(|Y - \hat{f}(X)| - \hat{s}(X) \leq \hat{t}_c|\mathcal{D}^{lrn}, \mathcal{D}^{cal}) \leq 1 - \alpha + \frac{1}{n_c + 1} + \sqrt{\frac{\log(1/\delta)}{n_c + 1}}. \tag{25}$$

See Section 2.1 and Proposition 24 in Humbert et al. (2024) for details on this result. Like in the proof of Theorem 3.7, we have $\mathbb{P}(\{(25)\} \cap \{(22)\} \cap \{(23)\}) \geq 1 - 3\delta$, and it suffices to show that if (25), (22) and (23) are true, we have our final upper-bound.

We have $\mathbb{E}\left[\lambda\left(C^{1-\alpha}_{\hat{f},\hat{s},\hat{t}_c}(X)\right)\Big|\mathcal{D}^{lrn}, \mathcal{D}^{cal}\right] = \mathbb{E}\left[\lambda\left(C^{1-\alpha}_{\hat{f},\hat{s},\hat{t}}(X)\right)\Big|\mathcal{D}^{lrn}\right] - 2\hat{t} + 2\hat{t}_c$. With (23) we have an upper-bound on $\mathbb{E}\left[\lambda\left(C^{1-\alpha}_{\hat{f},\hat{s},\hat{t}}(X)\right)\Big|\mathcal{D}^{lrn}\right]$, and it remains to show that $\hat{t}_c - \hat{t} \leq L\left(\frac{1}{(n_c+1)^\gamma} + 2\phi(\mathcal{F}, \delta, n_\ell)^\gamma\right)$.

Applying the quantile function $Q(\cdot; |Y - \hat{f}(X)| - \hat{s}(X)_{|\mathcal{D}^{lrn}})$ on (25), we get that $\hat{t}_c \leq Q(1 - \alpha + \frac{1}{n_c+1} + \sqrt{\frac{\log(1/\delta)}{n_c+1}}; |Y - \hat{f}(X)| - \hat{s}(X)_{|\mathcal{D}^{lrn}})$. Similarly, applying it on (22) gives $\hat{t} \geq Q(1 - \alpha - 2\phi(\mathcal{F}, \mathcal{S}, \delta, n_\ell); |Y - \hat{f}(X)| - \hat{s}(X))_{|\mathcal{D}^{lrn}}$. Hence, thanks to the regularity condition, we have:

$$\hat{t}_c - \hat{t} \leq L\Big(\frac{1}{n_c + 1} + \sqrt{\frac{\log(1/\delta)}{n_c + 1}} + 2\phi(\mathcal{F}, \mathcal{S}, \delta, n_\ell)\Big)^\gamma.$$

$\square$

## C. Detailed Implementation of the Empirical $(1 - \alpha)$-QAE Minimization

As explain in Section 3.3.1, to solve Problem (11) we use a gradient descent strategy. However, because the empirical quantile is not differentiable, we replace $\widehat{Q}$ in Problem (11) by the following smooth approximation:

$$\widetilde{Q}_\varepsilon(q; (\ell(\theta; Z_i))_{i \in \mathcal{D}^{lrn}}) = \inf\{t \ : \ \widetilde{F}_\varepsilon(t, \theta) \geq q\},$$

where $\widetilde{F}_\varepsilon$ is an approximation of the empirical distribution of the loss-values $(\ell(\theta; Z_i))_{i \in \mathcal{D}^{lrn}}$ defined for $\varepsilon > 0$ by

$$\widetilde{F}_\varepsilon(t, \theta) = \sum_{i \in \mathcal{D}^{lrn}} \Gamma_\varepsilon(\ell(\theta; Z_i) - t),$$

with

$$\Gamma_\varepsilon(z) = \begin{cases} 1 & z \leq -\varepsilon \\ \gamma_\varepsilon(z) & -\varepsilon < z < \varepsilon \\ 0 & z \geq \varepsilon \end{cases},$$

and $\gamma_\varepsilon : [-\varepsilon, \varepsilon] \longrightarrow [0, 1]$ a symmetric and strictly decreasing function such that it makes $\Gamma_\varepsilon$ differentiable. One possible choice for $\gamma_\varepsilon$ is given in (Pena-Ordieres et al., 2020, Eq. (2.6)):

$$\gamma_\varepsilon(z) = \frac{15}{16}\left(-\frac{1}{5}\left(\frac{z}{\varepsilon}\right)^5 + \frac{2}{3}\left(\frac{z}{\varepsilon}\right)^3 - \frac{z}{\varepsilon} + \frac{8}{15}\right). \tag{26}$$

For a given $q$ and $\varepsilon > 0$, under some assumptions on the loss (see Pena-Ordieres et al. (2020)), the implicit function theorem implies that:

$$\nabla_\theta[\widetilde{Q}_\varepsilon(q; (\ell(\theta; Z_i))_{i \in \mathcal{D}^{lrn}})] = \frac{\sum_{i \in \mathcal{D}^{lrn}} \Gamma'_\varepsilon(\ell(\theta; Z_i) - \widetilde{Q}_\varepsilon(q; (\ell(\theta; Z_i))_{i \in \mathcal{D}^{lrn}})) \cdot \nabla_\theta \ell(\theta; Z_i)}{\sum_{i \in \mathcal{D}^{lrn}} \Gamma'_\varepsilon(\ell(\theta; Z_i) - \widetilde{Q}_\varepsilon(q; (\ell(\theta; Z_i))_{i \in \mathcal{D}^{lrn}}))} , \quad (27)$$

where $\nabla_\theta$ denotes the gradient with respect to $\theta$ and $\Gamma'$ is the differential of $\Gamma$. We can therefore use a gradient descent algorithm to solve an approximation of the QAE Problem (11) given by:

$$\min_\theta \ \widetilde{Q}_\varepsilon(1 - \alpha; \{\ell(\theta; Z_i)\}_{i \in \mathcal{D}^{lrn}}) .$$

To this end, starting from an initial guess $\widetilde{\theta}_1$, we simply make the iterates:

$$\widetilde{\theta}_{k+1} = \widetilde{\theta}_k - \eta_k \nabla_\theta[\widetilde{Q}_\varepsilon(1 - \alpha; (\ell(\widetilde{\theta}_k; Z_i))_{i \in \mathcal{D}^{lrn}})] ,$$

where $\eta_k > 0$ is the step-size. The full procedure is summary in Algorithm 1 when $\gamma_\varepsilon$ is an in Eq. (26).

---

**Algorithm 1** Gradient descent to solve the QAE problem (step 1 of `EffOrt` and `Ad-EffOrt`)

---

1: **Inputs:** $\varepsilon, \widetilde{\theta}_1, n_{iter}, (\eta_k)_{1 \leq k \leq n_{iter}}, \alpha$
2: **for** $k = 1, \ldots, n_{iter}$ **do**
3:     $A \leftarrow \widetilde{Q}_\varepsilon(1 - \alpha; (\ell(\widetilde{\theta}_k; Z_i))_{i \in \mathcal{D}^{lrn}})$
4:     **for** $i \in \mathcal{D}^{lrn}$ **do**
5:         $B_i \leftarrow \Gamma'_\varepsilon(\ell(\widetilde{\theta}_k; Z_i) - A) = -\frac{15}{16} \left( \left( \varepsilon^2 - (\ell(\widetilde{\theta}_k; Z_i) - A)^2 \right)^2 / \varepsilon^5 \right) \cdot \mathbf{1}\{-\varepsilon < (\ell(\widetilde{\theta}_k; Z_i) - A) < \varepsilon\}$
6:         $C_i \leftarrow \nabla_\theta \ell(\theta; Z_i)$
7:     **end for**
8:     $\widetilde{\theta}_{k+1} \leftarrow \widetilde{\theta}_k - \eta_k \cdot \sum_i (B_i C_i) / \sum_i B_i$
9: **end for**
10: **Output:** $\widetilde{\theta}_{n_{iter}+1}$

---

*Remark* C.1. In our setting, $\ell$ is not differentiable because of the absolute value function. In practice, we therefore replace the gradient by a subdifferential (this is what we do in the experiments). Another possibility could be to replace the absolute value function with a smooth approximation, such as the Huber loss (Huber, 1964). Furthermore, as also done in Luo & Larson (2022), in Eq (27) we replace $\widetilde{Q}_\varepsilon(q; (\ell(\theta; Z_i))_{i \in \mathcal{D}^{lrn}})$ by the empirical quantile for computation efficiency.

*Remark* C.2. (Link with other formulations) Problem (11) is in fact similar to the *single chance constraint problem* (see e.g. (Curtis et al., 2018)). It can also be reformulated as the following bi-level optimization problem:

$$\min_\theta t(\theta) \quad \text{s.t.} \quad t(\theta) = \arg\min_t \sum_{i \in \mathcal{D}^{lrn}} \rho_{1-\alpha}(\ell(\theta; Z_i) - t) .$$

where $\rho_{1-\alpha}$ is the pinball loss. Indeed, from (Koenker & Bassett Jr, 1978; Biau & Patra, 2011) we know that $t(\theta) = \widehat{Q}(1 - \alpha; \{\ell(\theta; Z_i)\}_{i \in \mathcal{D}^{lrn}})$.

# D. Additional Results

## D.1. Synthetic Data

**Experimental setup details for Section 5.2:** During the learning step of `Ad-EffOrt`, we solve the $(1 - \alpha)$-QAE Problem (10) using the gradient descent strategy of Section 3.3.1. The smoothing parameter $\varepsilon$ is set to $0.1$, $n_{iter} = 1000$, and the step-size sequence is $\{(1/t)^{0.6}\}_{t=1}^{n_{iter}}$. The space of research $\mathcal{F}$ is restricted to the space of linear functions. The function $\hat{s}(\cdot)$ (second step of `Ad-EffOrt`) and the two quantile regression functions of CQR are learned by using a Random Forest (RF) quantile regressor, implemented in the Python package sklearn-quantile[1]. The function $\hat{\sigma}$ in LW-CP is learned using the RF regression implementation of scikit-learn (Pedregosa et al., 2011). Each time, the max-depth of the RF is set to $5$ and the other parameters are the default ones of the sklearn-quantile and scikit-learn packages.

---

[1] https://sklearn-quantile.readthedocs.io

**Additional experiments:** We now present additional results on synthetic data:

- In Figure 2, we display the coverage obtained on the scenarios of Section 5.1. We see that, as expected, all methods return sets with average coverage of $1 - \alpha = 0.9$ (white circle) regardless of the distribution of the noise.

- In figure 3, we present additional results obtained when the base predictor is a Networks (NNs) and not a linear regressor as made in the main paper. We consider the model $Y = X^2 + \mathcal{E}$ with $\mathcal{E}$ following the same distributions as presented in Section 5.1. In detail, we learn NNs with one hidden layer of size 10 and with a ReLU activation function. In `EffOrt`, the NN is learned using the gradient descent strategy of Section 3.3.1. The smoothing parameter $\varepsilon$ is set to 0.1, $n_{iter} = 1000$ and the step-size sequence is $\{(1/t)^{0.6}\}_{t=1}^{n_{iter}}$. The gradient with respect to the NN weights involved in the gradient descent is calculated using automatic differentiation. For split CP, the NN is learned using an ADAM optimizer and the loss is either a Huber loss (robust NN) or a least squares loss. Again, in all scenarios, `EffOrt` returns marginally valid sets in general smaller than those of the split CP method. This confirms that learning a model via the $(1 - \alpha)$-QAE problem is a better way of obtaining small prediction sets during the calibration step.

- In Figure 4, we display the coverage obtained on the scenarios of Section 5.2 when using `Ad-EffOrt`. We see again that, as expected, all methods return sets with average coverage of $1 - \alpha = 0.9$ (white circle) regardless of the distribution of the noise. Finally, Figure 5 shows examples of prediction sets returned by `Ad-EffOrt`, Locally weighted CP (LW-CP) and CQR when the noise is Gaussian.

**A different $\alpha$ between learning and calibration:** In our theoretical analysis, and in all the previous experiments, the $\alpha$ chosen to solve the QAE problem and the one used for the calibration step are the same. To highlight the impact of differently chosen values, we conduct additional experiments for `EffOrt`, with the linear regressor used in the main paper, when the QAE problem is solved with $\alpha_{QAE} = 0.05, 0.1$ and $0.9$ and the $\alpha$ of the calibration is $0.1$. Results are shown in Figure 6.

From Figure 6 top panels, we observe that QAE with $\alpha_{QAE} = 0.1$ produces the smallest prediction sets. However, when no extreme values are present, the choice of $\alpha_{QAE}$ appears to have little impact. In contrast, when extreme values exist in the distribution, selecting $\alpha_{QAE} = 0.05$ significantly deteriorates the final prediction sets. This is expected, as the dataset is constructed such that 5% of the values are extreme. Consequently, QAE with $\alpha_{QAE} = 0.05$ attempts to find $\hat{f}$ that minimizes the error for these extreme values, which is the opposite of being "robust." Figure 6 bottom panels confirm that the calibration step ensures the final coverage remains close to 0.9. Notice that a similar sensitivity to $\alpha$ can be observed in other conformal prediction methods, such as CQR, when the $\alpha_{CQR}$ values for the quantile regressors are poorly chosen.

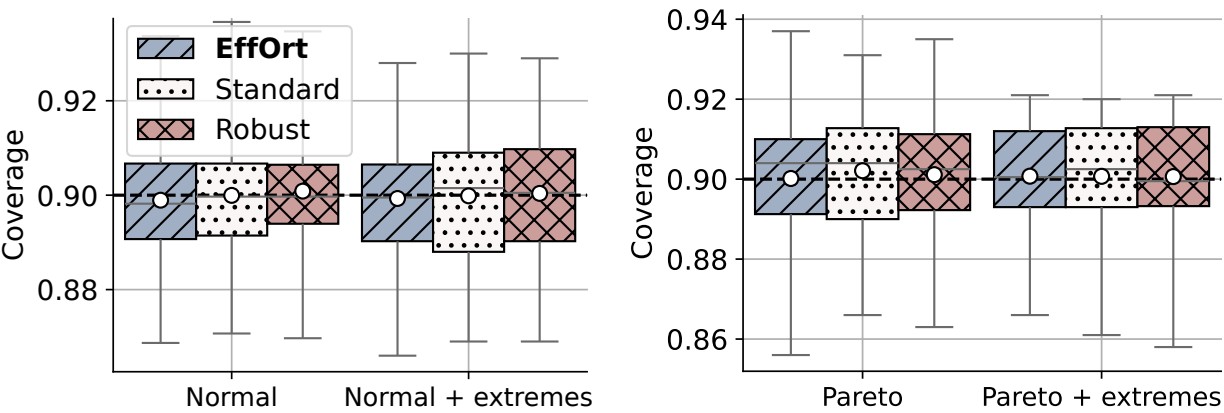

*Figure 2.* Synthetic data: Boxplots of the 50 empirical coverages obtained by evaluating `EffOrt` (see Section 5.1). The white circle corresponds to the mean.

### D.2. Real Data

We finally compare `Ad-EffOrt` with Locally Weighted CP (LW-CP) and CQR on the following public-domain real data sets also considered in e.g. (Romano et al., 2019): abalone (Nash et al., 1994), boston housing (housing) (Harrison Jr &

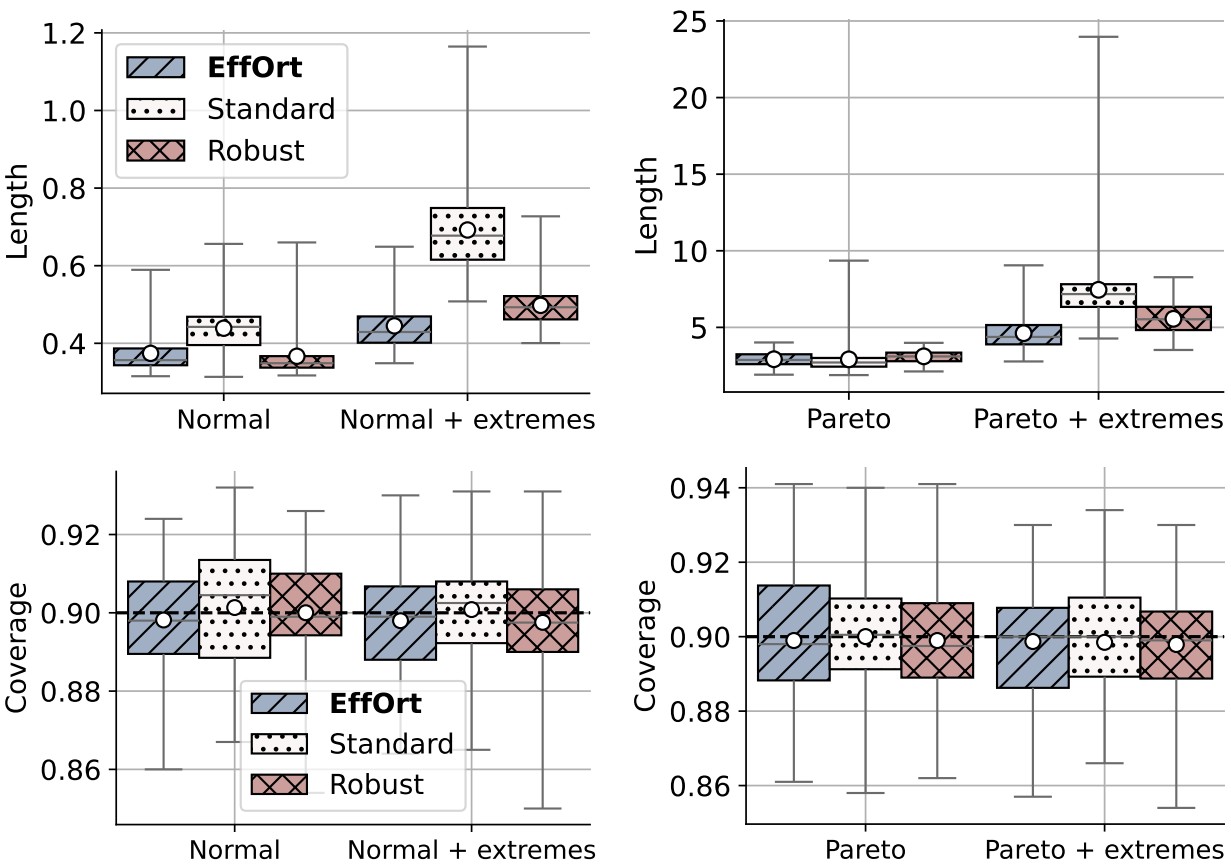

*Figure 3.* Synthetic data: Boxplots of the 50 empirical expected lengths (top) and coverages (bottom) obtained by evaluating `EffOrt` (see Section 5.1). The white circle corresponds to the mean.

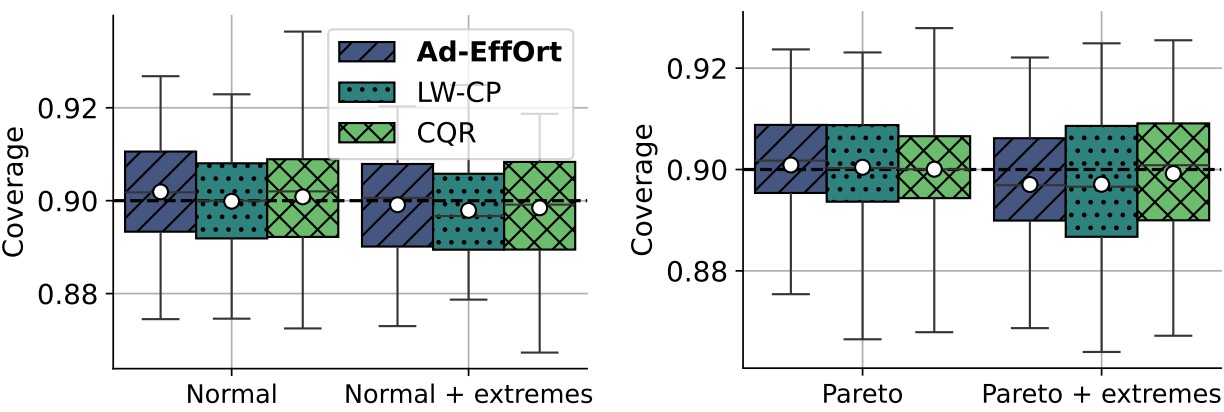

*Figure 4.* Synthetic data: Boxplots of the 50 empirical coverages obtained by evaluating `Ad-EffOrt` (see Section 5.2). The white circle corresponds to the mean.

Rubinfeld, 1978)[2], and concrete compressive strength (concrete) (Yeh, 1998).[3] We randomly split each data set 10 times into a training set, a calibration set and a test set of respective "size" 40%, 40%, and 20%. The training and calibration

---

[2]https://www.cs.toronto.edu/ delve/data/boston/bostonDetail.html
[3]http://archive.ics.uci.edu/dataset/165/concrete+compressive+strength

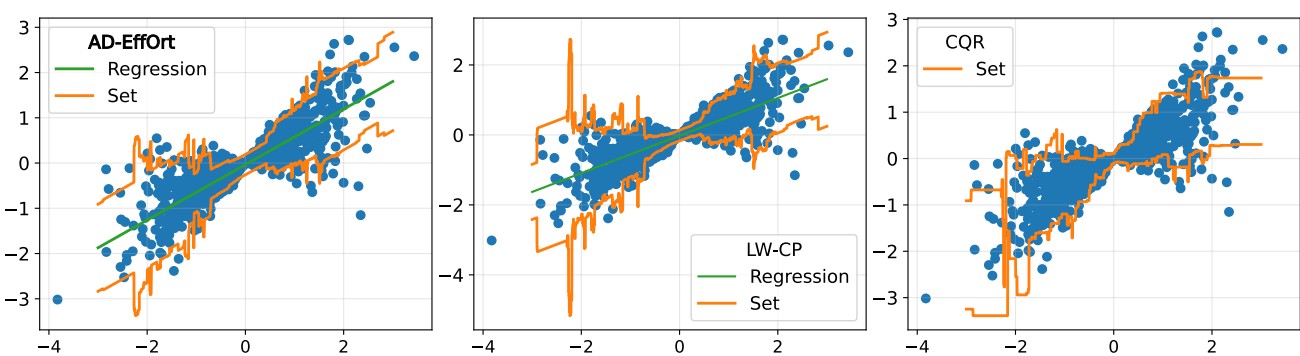

*Figure 5.* Synthetic data: Example of sets returned by `Ad-EffOrt` (left), LW-CP (middle), and CQR (right).

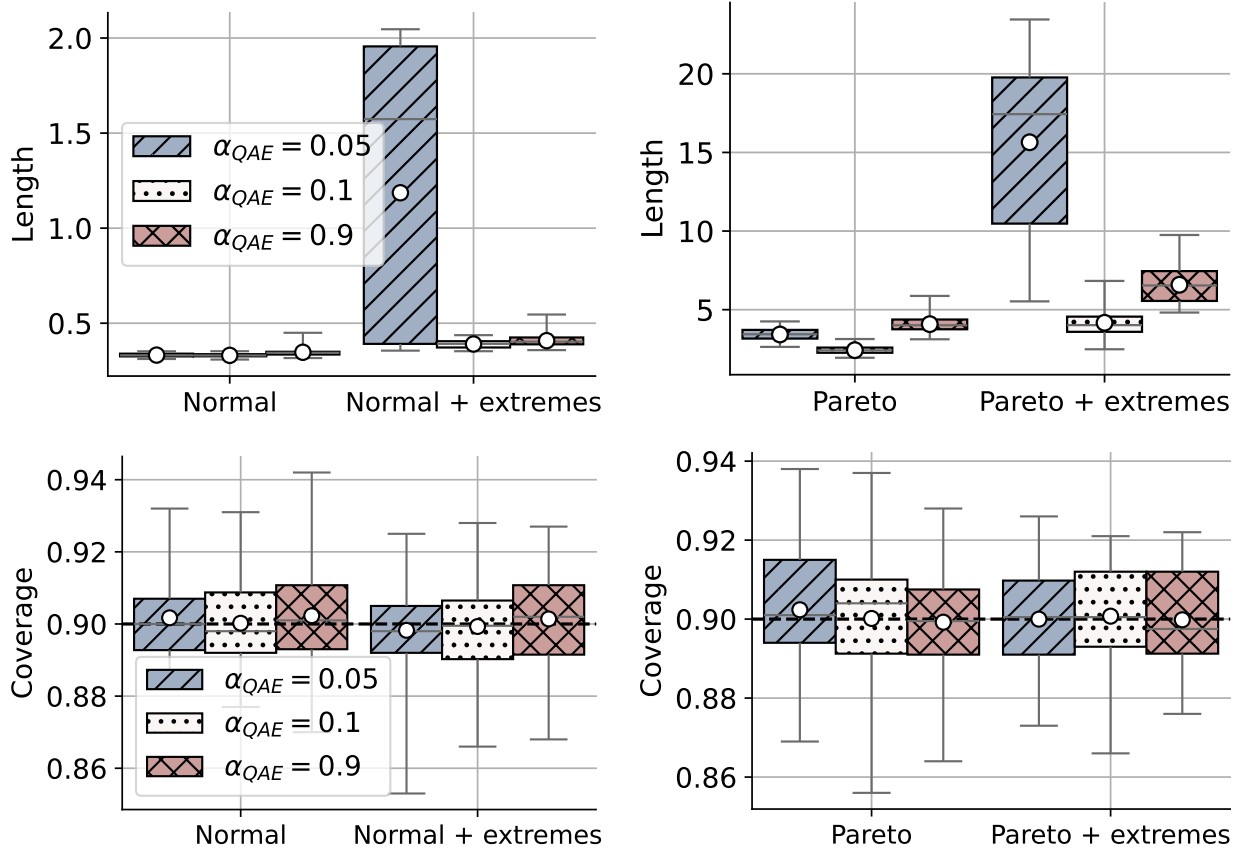

*Figure 6.* Synthetic data: Boxplots of the 50 empirical expected lengths (top) and coverages (bottom) obtained by evaluating `EffOrt` (see Section 5.1) for different values of $\alpha_{QAE}$ when solving the QAE problem. The white circle corresponds to the mean.

sets are used to apply `Ad-EffOrt`, LW-CP, and CQR, and the test set to compute the coverage and length metrics. For `Ad-EffOrt` and LW-CP the base prediction function $\hat{f}$ is a Neural-Network (NN) with one hidden layer of size 10 and a ReLU activation function. The function $\hat{s}$ in the step 2 of `Ad-EffOrt` and the two quantile regression functions of CQR are learned with a Random Forest (RF) quantile regressor, implemented in the Python package sklearn-quantile. The function $\hat{\sigma}$ in LW-CP is learned using the RF regression implementation of scikit-learn (Pedregosa et al., 2011). Each time, the max-depth of the RF is set to 5 and the other parameters are the default ones of the sklearn-quantile and scikit-learn packages. To illustrate the robustness of our approach, we finally add, in all the data sets, 5% of outliers to the values to be predicted, using a Gaussian distribution whose mean is equal to 2 times the maximum value of the original data.

Figure 7 displays the length and the normalized length (i.e. the length divided by the maximal length obtained with the three methods in the 10 splits) obtained on each data set. We can see that `Ad-EffOrt` is competitive, as it generally returns marginally valid sets (see figure 8 for coverage) of smaller or similar size to at least one of the other two methods. This is in line with the results obtained on synthetic data (Section 5 and Appendix D.1). Note also that the variability of the coverage metric (represented by the length of the boxes in Figure 8) is much smaller for `Ad-EffOrt` than LW-CP. Overall, these results show that `Ad-EffOrt` is empirically competitive with the main existing CP methods, while enjoying a strong theoretical grounding. It is therefore a method of choice for all practical applications.

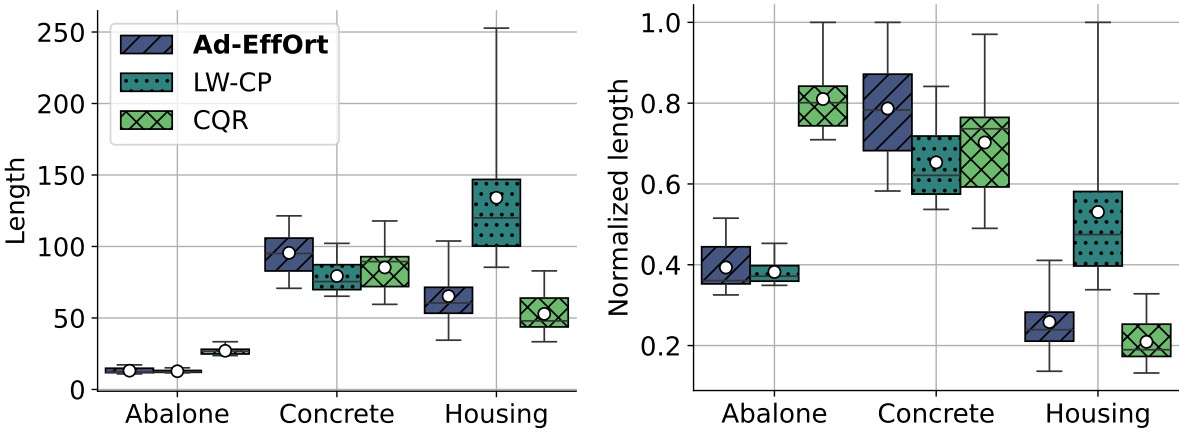

*Figure 7.* Real data: Boxplots of the lengths (left) and normalized lengths (right) obtained with `Ad-EffOrt`, LW-CP, and CQR on real data sets. The white circle corresponds to the mean.

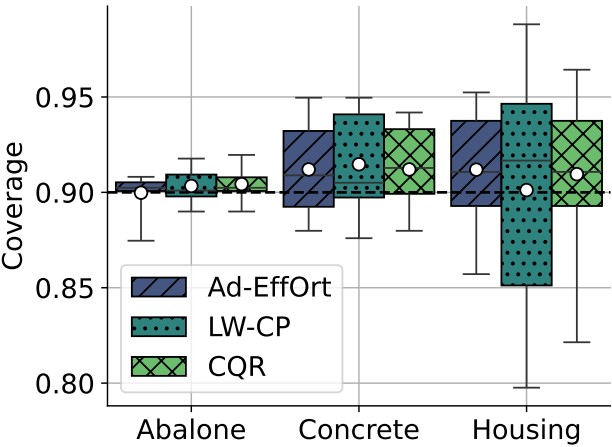

*Figure 8.* Real data: Boxplots of the coverages obtained with `Ad-EffOrt`, LW-CP, and CQR on real data sets. The white circle corresponds to the mean.

