# OpenReview forum: "On Volume Minimization in Conformal Regression"
_ICML.cc/2025/Conference — ICML 2025 poster_

### Official Review · Reviewer_jRxu · 2025-02-20

**Overall Recommendation:** 4

**Summary:**

The authors present a method that minimizes the volume of a conformal prediction region, subject to the coverage being the desired one (1- alpha). They show its theoretical properties, and its empirical validity.

**Claims And Evidence:**

The claims are clear and convincing. I'd like the authors, though, to address all my questions (see below).

**Essential References Not Discussed:**

Some references to known problems and results are missing. Please refer to the Questions section.

**Experimental Designs Or Analyses:**

The experimental designs and analyses seem sound and valid.

**Methods And Evaluation Criteria:**

The proposed evaluation criteria are apropos for the authors' goal.

**Other Comments Or Suggestions:**

Please refer to the Questions section.

**Other Strengths And Weaknesses:**

The paper is well-written, and studies an interesting problem. I gave it a score of 3, with the willingness to increase it if the authors are able to address the questions I raise.

**Questions For Authors:**

**Q1** In https://link.springer.com/book/10.1007/b106715, Theorem 2.10, the authors provide a proof by construction, that can be used to derive the narrowest possible (in the sense of diameter or volume) conformal prediction region. How does the authors' work relate to that result?

**Q2** In https://proceedings.mlr.press/v216/sale23a.html, the authors show that the volume is not a good measure for (epistemic) uncertainty for classification problems with more than 2 classes. How does this result compare with what the authors present in their paper?

**Q3** Despite the ubiquitous claim that conformal prediction (CP) is an uncertainty *quantification* tool, it is actually not. CP is an uncertainty *representation* tool. Indeed, CP *represents* uncertainty via the conformal prediction region. It does not quantify it: there is no real value attached to any kind of predictive uncertainty (aleatoric or epistemic, AU and EU, respectively). Some claim that the diameter (or the volume, like in the authors' work) of the conformal prediction region quantifies the uncertainty, but even in that case, it is unable to distinguish between AU and EU. Indeed, the diameter (and the volume too) is a positive function of both: it increases as both increase, and hence it cannot be used to distinguish between the two. This was already pointed out in https://openreview.net/forum?id=4NHF9AC5ui. I'd appreciate a discussion by the authors on this matter. I also briefly mention that a recent paper merged CP with imprecise probabilistic tools to obtain a "proper" AU and EU disentanglement https://openreview.net/forum?id=L7sQ8CW2FY. The authors may find it interesting, and possibly useful for the discussion section and/or for future research.

**Q4** It is also worth to notice that CP is not entirely model-free, since the volume (and diameter) of the resulting conformal prediction region depends on the choice of the non-conformity score. This was already pointed out in https://www.arxiv.org/abs/2502.06331. The authors are suggested to include this fact in their work.

**Relation To Broader Scientific Literature:**

The paper advances the state of the art on conformal prediction region methods, even though some related works are not considered. Please refer to the Questions section.

**Theoretical Claims:**

The theoretical claims seem largely correct.

---

> ### Author Rebuttal · Authors · 2025-03-31
>
> We sincerely thank the reviewer for their valuable feedback and for pointing out relevant related work. Below, we address each of their questions and comments.
>
>
> - *Q1 In https://link.springer.com/book/10.1007/b106715, Theorem 2.10, the authors provide a proof by construction, that can be used to derive the narrowest possible (in the sense of diameter or volume) conformal prediction region. How does the authors' work relate to that result?.*
>
> We appreciate the reviewer for highlighting this result. After carefully examining Theorem 2.10 and its proof, we find that our work does not explicitly relate to it. From our understanding, the proof constructs a theoretical conformal set that is guaranteed to be at least as efficient as a given conservatively valid confidence predictor. However, it is not evident that this result can be leveraged to construct a conformal predictor of minimal size in practice. In contrast, our approach explicitly formulates and empirically solves the length minimization problem.
>
> - *Q2 In https://proceedings.mlr.press/v216/sale23a.html, the authors show that the volume is not a good measure for (epistemic) uncertainty for classification problems with more than 2 classes. How does this result compare with what the authors present in their paper?.*
>
> The referenced paper differs significantly from our work. First, it does not explicitly address Conformal Prediction (CP) but rather focuses on credal sets as a means of quantifying uncertainty. Additionally, their analysis is centered on classification, whereas our work focuses on regression (extending to high-dimensional settings is left as future work by the authors). That being said, the authors do show that, in dimensions greater than two, measuring uncertainty through the volume of the credal set is inadequate—at least with respect to the axioms they propose (Section 3 of [1]). To draw a direct comparison between our work and the one mentioned by the reviewer, one would need to establish a precise connection between credal sets and conformal prediction and investigate whether the results in [1] extend to the volume of conformal prediction sets. We find this to be an interesting open question for further analysis of CP as an Uncertainty Quantification (UQ) technique.
>
> [1] https://proceedings.mlr.press/v216/sale23a.html
>
> - *Q3 Despite the ubiquitous claim that [...] for future research.*
>
> We thank the reviewer for bringing these references to our attention. We fully agree that CP is better described as an uncertainty representation tool rather than a UQ technique, which is why we deliberately avoided making such a claim in our work. It appears that this comment, like the previous question, highlights a general limitation of CP rather than a specific limitation of our contribution. Nevertheless, we find that linking CP with more classical UQ frameworks (e.g., credal sets) presents an exciting research direction. We will mention this in the conclusion of our paper as a promising avenue for future work.
>
> - *Q4 It is also worth to notice that CP is not entirely model-free, since the volume (and diameter) of the resulting conformal prediction region depends on the choice of the non-conformity score. This was already pointed out in https://www.arxiv.org/abs/2502.06331. The authors are suggested to include this fact in their work.*
>
> We fully agree with the reviewer, as this observation aligns with one of the conclusions of our paper: the choice of the prediction set class $\mathcal{C}$, which depends on $\mathcal{F}$ in Section 3, has a significant impact on the volume of the conformal prediction region. We will clarify this point in the final version of our manuscript and include a reference to the mentioned paper (noting that it was made available after the ICML submission deadline).

---

### Official Review · Reviewer_d77L · 2025-02-22

**Overall Recommendation:** 3

**Summary:**

The paper theoretically studies the efficiency of CP sets, providing bounds for the specific task of regression. The bounds are given assuming a fixed base predictor, or in the case where the predictor is learned on held-out data (split CP). For the latter case, the authors also highlight the importance of minimizing the Quantile Absolute Error (QAE) in order to reduce inefficiency at calibration time. This insight is then used to propose two algorithms, EffOrt and Ad-EffOrt, which minimize a surrogate of the QAE. The solution is tested for regression tasks.

**Claims And Evidence:**

The paper claims to contribute to the existing literature by deriving upper bounds on the size of split CP and proposing a training step for split CP aimed at minimizing the set size of prediction sets.

**Essential References Not Discussed:**

I believe that the current literature review misses many existing works.

To the best of my knowledge, the first paper to theoretically study the expected set size of CP sets was [1]. This paper studies the set size for different tasks, including the regression task considered here. It examines the expected size under a fixed predictor. This analysis has been extended to split CP, where the learner is not fixed but learned, both under the i.i.d. setting and covariate shift [2-3]. While these works have not proposed algorithmic solutions to minimize the expected inefficiency of CP sets, [4] and [5] proposes a training algorithm starting from the PAC-Bayes bound on the efficiency of split CP and information-theoretic tools.

[1]. Dhillon, Guneet S., George Deligiannidis, and Tom Rainforth. "On the expected size of conformal prediction sets." International Conference on Artificial Intelligence and Statistics. PMLR, 2024.
[2] Zecchin, Matteo, et al. "Generalization and informativeness of conformal prediction." 2024 IEEE International Symposium on Information Theory (ISIT). IEEE, 2024.
[3] Zecchin, Matteo, et al. "Generalization and Informativeness of Weighted Conformal Risk Control Under Covariate Shift." arXiv preprint arXiv:2501.11413 (2025).
[4] Sharma, Apoorva, et al. "PAC-Bayes generalization certificates for learned inductive conformal prediction." Advances in Neural Information Processing Systems 36 (2023): 46807-46829
[5] Correia, Alvaro, et al. "An information theoretic perspective on conformal prediction." Advances in Neural Information Processing Systems 37 (2025): 101000-101041.

**Experimental Designs Or Analyses:**

Experiments are carried out on simple synthetic data; however, the Appendix contains more extended evaluation.

**Methods And Evaluation Criteria:**

The problem is tested on synthetic data; it would be better to test it on actual benchmarks used in CP.

**Other Comments Or Suggestions:**

None

**Other Strengths And Weaknesses:**

I think the paper does a nice job of leveraging insights from theoretical results to propose a new training algorithm. The proposed methods seem to have advantages over existing ones. However, the paper overlooks many existing works, which I believe undermines its originality claims and experimental results. For example, how would the algorithms in [4]-[5] compare to EffOrt?

From my understanding, when using EffOrt, one commits to a coverage level $\alpha$, assuming that the same \alpha will be used for calibration. However, standard CP is a post-hoc calibration method, and the training of the base predictor is independent of $\alpha$. I suggest including experiments to evaluate the performance of EffOrt when $\alpha$ at calibration time differs from the one used for training. In many instances, retraining a model is not feasible.

**Questions For Authors:**

None

**Relation To Broader Scientific Literature:**

There is a quite a lot of missing literature. There already exists works that study the expected set size or informativeness of split CP in standard settings as well as under covariate shift.

**Theoretical Claims:**

I haven't carefully checked the proofs but the results are coherent with existing ones.

---

> ### Author Rebuttal · Authors · 2025-03-31
>
> We sincerely appreciate the reviewer’s thoughtful feedback and the references provided. We acknowledge that some relevant prior works were overlooked, and we will incorporate them into the related work section to ensure a more comprehensive discussion. While we recognize that this may slightly refine the novelty claims of certain aspects of our work, we firmly believe that our contribution remains **distinct and complementary** to these existing efforts.
>
> In particular, while most of the cited works (except [5], which focuses on linking CP set size to the conditional entropy $H(Y|X)$) aim to theoretically analyze the size of conformal prediction (CP) sets, **none explicitly study the notion of "excess volume loss"** introduced in our work. This metric quantifies the gap between the CP set size and an oracle reference, offering a more precise measure of inefficiency. For instance, while [4] provides a PAC-Bayesian upper bound on CP set size, their analysis does not incorporate an oracle baseline, preventing an explicit characterization of suboptimality.
>
> Below, we present a new paragraph that will be added to the related work section, directly addressing the reviewer’s concerns.
>
> ---
>
>  **New Paragraph: Comparison with Missing Related Works**
>
> *Recent studies [1,2,3] have analyzed the expected size of split CP sets, highlighting key factors such as the impact of the score function choice [1] and the generalization properties of the base predictor [2,3]. In [4], the authors take a step further by deriving a PAC-Bayesian upper bound on the expected CP set size, involving an empirical estimate of the size and a KL divergence term. They also propose an algorithm that modifies the calibration step of split CP to minimize their bound; however, they do not explicitly instantiate their bound for their proposed algorithm, leaving the practical implications unclear.*
>
> *Our work differs from these approaches in several fundamental ways. First, our theoretical guarantees are PAC-based rather than PAC-Bayesian, eliminating the need for restrictive assumptions such as boundedness of the score function or target variable $Y$. More crucially, our analysis introduces an upper-bound on the excess volume loss, explicitly quantifying how much larger the CP set is compared to an oracle reference. This perspective is absent in prior works, which focus only on bounding the expected CP set size without reference to an optimal baseline. Finally, it is worth mentioning [5], which provides an information-theoretic perspective by linking CP set size to conditional entropy.*
>
> ---
>
> **$\alpha$ different between learning and calibration**
>
>
> This is an interesting question. To address it, we conducted additional experiments where the QAE problem was solved with $\alpha_{QAE} = 0.05, 0.1$ and $0.9$. The results are presented in the following anonymous figures:
> (1) [Figure 1](https://ibb.co/fYR6MDgy) – Length for Normal and Normal + Extreme,
> (2) [Figure 2](https://ibb.co/5gdtRJ0t) – Length for Pareto and Pareto + Extreme,
> (3) [Figure 3](https://ibb.co/HTLb2DS4) – Coverage for Normal and Normal + Extreme,
> (4) [Figure 4](https://ibb.co/S488nwK6) – Coverage for Pareto and Pareto + Extreme.
>
> From Figures 1 and 2, we observe that QAE with $\alpha_{QAE}=0.1$ produces the smallest prediction sets. However, when no extreme values are present, the choice of $\alpha_{QAE}$ appears to have little impact. In contrast, when extreme values exist in the distribution, selecting $\alpha_{QAE}=0.05$ significantly deteriorates the final prediction sets. This is expected, as the dataset is constructed such that 5% of the values are extreme. Consequently, QAE with $\alpha_{QAE} = 0.05$ attempts to find $\hat{f}$ that minimizes the error for these extreme values, which is the opposite of being "robust." Figures 3 and 4 confirm that the calibration step ensures the final coverage remains close to 0.9.
>
> Notice that a similar sensitivity to $\alpha$ can be observed in other conformal prediction methods, such as CQR, when the $\alpha_{CQR}$ values for the quantile regressors are poorly chosen.

---

> > ### Comment · Reviewer_d77L · 2025-04-03
> >
> > Thanks for acknowledging the existence of these past works. I also believe there is a degree of orthogonality between previous research and the results presented in the paper. The paragraph you included does a good job of clarifying these aspects.
> >
> > I also appreciate these new results; to me, they look very insightful and worth including in the final version—perhaps in the appendix.
> >
> > I will raise my score in light of these modifications.

---

### Official Review · Reviewer_J4Qu · 2025-02-25

**Overall Recommendation:** 4

**Summary:**

**Post-rebuttal edit:** The authors wrote convincing answers to my two grounds and questions. I keep my positive score.

&nbsp;

The submission considers (split) conformal prediction for univariate data, with scores given by absolute values of the residuals, and with prediction regions given by intervals of the form $C(x) = \hat{f}(x) - \hat{t_\alpha}, \hat{f}(x) + \hat{t_\alpha}$ in the main case (Section 3), where $\hat{f}$ is learnt on the learning set and where the $\hat{t}_\alpha$ are learnt on the calibration set, as quantiles of the empirical distribution of scores. The coverage level of these $C(x)$ are approximately $1-\alpha$, under an exchangeability condition, as follows from a well-known analysis. The regression functions belong to some set $\mathcal{F}$. The focus of this article is to relate the length of $C(x)$ to the length obtained by using the ground-truth quantile of the underlying distributions (depicted as the ideal length) and picking the optimal element in $\mathcal{F}$ in this respect.

The main results are the final problem statement in Section 3.2 and the upper bound on the interval length given by Theorem 3.7 in Section 3.4: under natural assumptions (regularity of the quantile function, bound on the complexity of $\mathcal{F}$), the length of the $C(x)$ is smaller than the ideal length plus two closed-form terms, both converging to 0 as, respectively, the sizes of the training and calibration sets increase (and where the term due to the learning set is larger; thus the learning set should feature more data points than the calibration set).

These main results are stated for an ideal version of the procedure (with a computational bottleneck); but simulations report effective results for a computationally more tractable version of the procedure. Also, an extension (without theoretical results) to intervals with lengths depending on the features $x$ is considered.

**Claims And Evidence:**

Yes, as far as the setting and the results of Sections 3-4-5 are concerned; see the detailed discussions below, both for theoretical claims and simulations.

**Essential References Not Discussed:**

No issue spotted.

**Experimental Designs Or Analyses:**

I read Section 5, which looks to me like a decent set of experiments on artificial data.

**Methods And Evaluation Criteria:**

The setting considered makes sense.

**Other Comments Or Suggestions:**

Sections 1 and 2 could perhaps focus faster on the specific class of problems studied in Section 3; they take quite some space to discuss the problem in generality, but the problem is later not studied at this degree of generality. Also, results like the rewriting of the optimization problem as Equation (10) could fit earlier in the article.

In particular, in Section 2.2, the optimization problem (4) is stated in great generality whereas the optimization problem (5) corresponds to a specific (yet legitimate) choice of non-conformity score. This dependence shoud have been emphasized, as in practice, the choice of the non-conformity score usually has a larger impact on the efficiency than the loss used in the training step.

Section 5.1: Could you detail what a 'robust linear regression with Huber loss' is, and why it helps for a fair comparison?

In Section 5, you take $n_{lrn} = n_{cal} = 1000$. Are these values large enough for the conditions in Theorem 3.7 to be met?

Typos
- 044:1 "miscoverage" instead of "coverage"
- 055:2 drop the "for any $x$"
- 403:2 "coverages" instead of "coverage"
- 699 "thanks to (20)"

Suggestions
- State Assumption 3.5 and Proposition 3.6 with a $n_l$ instead of a $n$?

**Other Strengths And Weaknesses:**

The problem of interval-length-minimization is natural in (split) conformal prediction, yet, few results were available. This submission provides an elegant approach for this: the additional length due to calibration is handled through the DKW inequality (on empirical cdfs), while the additional length due to learning is bounded through the theory of suprema of empirical processes (hidden in Assumption 3.5). These classic tools of statistical learning are well used in the context of split conformal prediction. This is the first strength of the submission.

The other strength is the clarity of the exposition and writing. In particular, it was a nice idea to discuss first the error due to calibration (in Section 3.3) and add later (in Sections 3.2 and 3.4) the error due to learning. Also, the final paragraph of Section 3.4, about the respective importances of the train and calibration sets, is an important take-home message.

**Questions For Authors:**

Q1. The assumptions to obtain Theorem 3.7 are not clearly stated: are the training and the calibration data assumed to be i.i.d.?

Q2. I guess that unfortunately, there are no theoretical guarantees for the computationally more efficient procedure described in Section 3.3? This looks like a critical limitation, which my score takes into account (I would raise it if I misunderstood this).

Q3. Could you clarify the statement "theoretical analysis highlights how the complexity of the prediction function classes impacts the prediction interval's length"? From my understanding the upper bound (13) given in Theorem 3.7 is only an upper bound, and can only give some insights, especially as both its first and third terms depend on the complexity of the prediction function classes.

Q4. Did you consider the length minimization problem for another non-conformity score? If you did, do you know if similar results hold for signed non-conformity scores (such as the residuals), which seem more adapted to the asymmetric heavy-tailed distribution you considered in Section 5?

Q5. I am surprised that the empirical coverages given in Figures 2-3-4 are, on average, slightly below the nominal coverage of 90\%. How did you implement the SCP procedure in your experiments?

&nbsp;

**Post-rebuttal edit:** The authors wrote convincing answers to my two grounds and questions. I keep my positive score.

**Relation To Broader Scientific Literature:**

The submission discusses well the broader literature, like the minimum volume estimation problem, typically dealt with through density estimation.

**Theoretical Claims:**

I checked in details the proofs of Proposition 3.1 and Theorem 3.7 (located in Appendix A), and could spot no issue.

---

> ### Author Rebuttal · Authors · 2025-03-31
>
> We sincerely thank the reviewer for their thoughtful and constructive feedback. We are especially grateful for their recognition of the clarity and novelty of our contribution, particularly our use of statistical learning theory in the context of split conformal prediction. We will incorporate the suggested writing improvements in the final version of the manuscript. Below, we address the reviewer’s main questions and remarks.
>
> **Questions**
>
> - *Q1. The assumptions to obtain Theorem 3.7 are not clearly stated: are the training and the calibration data assumed to be i.i.d.?*
>
> Yes, our theoretical analysis, including Theorem 3.7, assumes that both the training and calibration data are i.i.d. We will explicitly state this assumption in the manuscript.
>
> - *Q2. I guess that unfortunately, there are no theoretical guarantees for the computationally more efficient procedure described in Section 3.3? This looks like a critical limitation, which my score takes into account (I would raise it if I misunderstood this).*
>
> The reviewer is correct—our theoretical guarantees currently apply only to the idealized procedure, and we do not yet have formal guarantees for the computationally efficient variant. We acknowledge this as a limitation and appreciate the reviewer bringing it to our attention. We will explicitly discuss this in the manuscript and highlight it as an important direction for future theoretical research.
>
> - *Q3. Could you clarify the statement "theoretical analysis highlights how the complexity of the prediction function classes impacts the prediction interval's length"? From my understanding the upper bound (13) given in Theorem 3.7 is only an upper bound, and can only give some insights, especially as both its first and third terms depend on the complexity of the prediction function classes.*
>
> The reviewer is absolutely right that Theorem 3.7 provides an upper bound rather than a precise characterization of the true interval length. Our goal was to emphasize that this bound offers insights into how the complexity of the function class influences the gap between the constructed interval length, $\lambda(C_{\hat{f},\hat{t}}^{1-\alpha})$, and the optimal one $\lambda(C_{f^*,t^*}^{1-\alpha})$. Specifically, as the complexity of the function class increases, more training data is required for the bound on this difference to converge.
>
> - *Q4. Did you consider the length minimization problem for another non-conformity score? If you did, do you know if similar results hold for signed non-conformity scores (such as the residuals), which seem more adapted to the asymmetric heavy-tailed distribution you considered in Section 5?*
>
> To clarify, our framework does not explicitly rely on a chosen non-conformity score. Instead, the choice of score function is implicitly determined by the class of prediction sets, $\mathcal{C}$, that we consider—namely, intervals of constant (Section 3) or adaptive (Section 4) lengths. Conversely, in standard split conformal prediction, one can view the choice of non-conformity score as implicitly defining a class of prediction sets.
>
> In Section 3, the scores used in the calibration step correspond to absolute residuals $|y-f(x)|$, whereas in Section 4, they take the form $|y-f(x)| - s(x)$, which are signed scores. To answer the reviewer’s question, since we have not considered alternative classes of prediction sets, we have not explored other types of score functions. However, as emphasized in our conclusion, extending our framework to other classes of prediction sets (and therefore score functions) is an important direction for future work.
>
> - *Q5. I am surprised that the empirical coverages given in Figures 2-3-4 are, on average, slightly below the nominal coverage of 90%. How did you implement the SCP procedure in your experiments?*
>
> We appreciate the reviewer pointing out this observation. After re-running our experiments, we found that the slight undercoverage was due to the limited number of repetitions in our empirical evaluation—averaging over 50 trials was not sufficient to smooth out random fluctuations. Indeed, due to the inherent stochasticity of different random datasets, finite-sample variability can cause slight deviations from the coverage. We will clarify this in the appendix.

---

> > ### Comment · Reviewer_J4Qu · 2025-04-02
> >
> > **RE Q4:** I'm unsure if I'm getting your point. The class of prediction sets you consider in equation (5) line 171 does not include all the constant-length intervals and is, I believe, limited to using the absolute residuals. My question was about considering other intervals of constant lengths, produced by considering the quantiles $\alpha/2$ and $1-\alpha/2$ of the calibration residuals as the lower and upper bounds of the intervals (see for example Linusson et al. [2014], Signed-error conformal regression). In this configuration, the length of the interval is no longer $2t$, and I wonder if you think that a similar analysis can be derived in this case.

---

> > > ### Author Response · Authors · 2025-04-07
> > >
> > > We thank the Reviewer for the clarification and for raising this interesting question. If we understand correctly, the suggestion is to consider an extension of our framework to a broader class of set-valued functions, parameterized by  $a \leq b \in \mathbb{R}$ and $f \in \mathcal{F}$, such that:
> > >  $C_{f,a,b}: x \mapsto [f(x) + a, f(x) + b]$, with a fixed length $b - a$. This class notably includes the prediction sets proposed in Linusson et al. [2014].
> > >
> > > Unfortunately, **when $f$ is fixed**, deriving a closed-form expression for the optimal values of $a$ and $b$ becomes challenging. One of the main difficulties lies in the fact that this family is not nested (as discussed in Appendix B.1) and that optimal values of $a$ and $b$ depend on each other, which prevents us from expressing them directly as a function of the score distribution. Note also that while choosing $a$ and $b$ as the $\alpha/2$ and $1 - \alpha/2$ quantiles of the scores (see Linusson et al. [2014]) would ensure valid coverage, such a choice does not necessarily minimize the length of the prediction interval, especially when the distribution of the scores is asymmetric.
> > >
> > > That said, an important observation is that **when $f$ is not fixed**, the above class of set-valued functions actually coincides with the class studied in our paper (denoted $\mathcal{C}^{\text{const}}_\mathcal{F}$), up to a mild assumption on the function class $\mathcal{F}$. Specifically, if $\mathcal{F}$ is stable under scalar translation (i.e., for all $f \in \mathcal{F}$, $f + k \in \mathcal{F}$ for any $k \in \mathbb{R}$—a condition satisfied by most standard models that include an intercept term), then the set $C$ parametrized by $a,b,f$ belongs to the class considered in our work. To see this more concretely, one can take $f' = f + (a + b)/2$ and $t = (b - a)/2$. This recasts the interval $[f(x) + a, f(x) + b]$ as $[f'(x) - t, f'(x) + t]$, which is exactly the form we analyze in our work. As a result, our theoretical analysis and results continue to apply in this setting.

---

### Official Review · Reviewer_gwDX · 2025-03-17

**Overall Recommendation:** 3

**Summary:**

Conformal prediction is a framework to construct label sets such that the marginal probability of coverage is guaranteed to be above a desired level $(1 - \alpha) \in (0, 1)$. This paper studies the conformal label intervals (one contiguous set) for unidimensional regression problems. The motivation is to minimize the marginal interval width (inefficiency) while maintaining the conformal coverage guarantee.

In doing so, the paper first considers fixed-width label intervals. Under this setting, the optimal regressor that predicts the center of the label interval is the one that minimizes the $(1 - \alpha)$-th quantile of the residual distribution. The paper proposes EffOrt, a regressor trained to minimize an empirical approximation of the same. Additionally, the paper derives a bound on the difference in interval widths between the proposed method and an oracle, where the bound depends on the number of training and calibration data and the complexity of the function class.

The paper also proposes to learn to predict the interval widths to allow adaptivity. The optimal regressor is the one that minimizes the $(1 - \alpha)$-th quantile of the conditional residual distribution. The paper proposes Ad-EffOrt, a conditional quantile regressor atop EffOrt, to approximate an empirical approximation of the same.

**Claims And Evidence:**

Yes, the claims are supported.

**Essential References Not Discussed:**

There are works on the marginal size of the conformal label sets, termed inefficiency. Most show that conformal inefficiency asymptotically converges to that of an oracle under different settings: unsupervised learning [Lei et al., 2013, 2015], regression [Lei and Wasserman, 2013], binary classification [Lei, 2014], and multi-class classification [Sadinle et al., 2019]. Similarly, Vovk et al. [2014, 2016] and Sadinle et al. [2019] provide results under per-class/label coverage. Additionally, Dhillon et al. [2024] quantify conformal inefficiency in the finite-sample setting.

The paper includes some but not all references.

References

G. S. Dhillon, G. Deligiannidis, and T. Rainforth. On the expected size of conformal prediction sets. In S. Dasgupta, S. Mandt, and Y. Li, editors, Proceedings of The 27th International Conference on Artificial Intelligence and Statistics, volume 238 of Proceedings of Machine Learning Research, pages 1549–1557. PMLR, 02–04 May 2024.

J. Lei. Classification with confidence. Biometrika, 101(4):755–769, 10 2014.

J. Lei and L. Wasserman. Distribution-free prediction bands for non-parametric regression. Journal of the Royal Statistical Society Series B: Statistical Methodology, 76(1):71–96, 07 2013.

J. Lei, J. Robins, and L. Wasserman. Distribution-free prediction sets. Journal of the American Statistical Association, 108(501):278–287, 2013.

J. Lei, A. Rinaldo, and L. Wasserman. A conformal prediction approach to
explore functional data. Annals of Mathematics and Artificial Intelligence, 74(1):29–43, Jun 2015.

M. Sadinle, J. Lei, and L. Wasserman. Least ambiguous set-valued classifiers with bounded error levels. Journal of the American Statistical Association, 114(525):223–234, 2019.

V. Vovk, I. Petej, and V. Fedorova. From conformal to probabilistic prediction. In L. Iliadis, I. Maglogiannis, H. Papadopoulos, S. Sioutas, and C. Makris, editors, Artificial Intelligence Applications and Innovations, pages 221–230, Berlin, Heidelberg, 2014.

V. Vovk, V. Fedorova, I. Nouretdinov, and A. Gammerman. Criteria of efficiency for conformal prediction. In A. Gammerman, Z. Luo, J. Vega, and V. Vovk, editors, Conformal and Probabilistic Prediction with Applications, pages 23–39, Cham, 2016.

**Experimental Designs Or Analyses:**

Yes, the experimental design is sound. However, during the analysis, the paper highlights that Ad-EffOrt constructs label sets with more consistent interval widths (compared to the other baselines). This property is not necessarily beneficial because varied interval widths can signify adaptivity and help achieve (approximate) conditional coverage.

**Methods And Evaluation Criteria:**

Yes, they make sense. However, an additional baseline would be the methods by Stutz et al. [2022] and Kiyani et al. [2024]. They minimize the marginal label set size (an empirical approximation of it) with approximations and derivations similar to the ones in this paper. For instance, approximating the indicator function, minimizing the $(1 - \alpha)$-th quantile of the residual distribution, etc.

**Other Comments Or Suggestions:**

1.	Including the experimental results on real-world data in the main paper, over the synthetic ones, will help.

2.	Including a brief description of approximating the indicator function in the main paper will help.

3.	The term "the space of research" is used many times but does not sound correct. The "search space" or "function class" might be better.

Typos:

1.	"This would allows deriving..." $\rightarrow$ "This would allow deriving..." (line 364, column 1).

2.	"...while $s$ in learned in order..." $\rightarrow$ "...while $s$ is learned in order..." (line 368-369, column 2).

3.	"...both empirically showed to be..." $\rightarrow$ "...both empirically shown to be..." (line 422-423, column 2)

**Other Strengths And Weaknesses:**

Strengths:

1.	The paper is well-written and easy to follow. The ideas progress well, making the function class progressively more flexible.

2.	The area of research is well-motivated for practical impact.

Weaknesses:

1.	The analysis and techniques used are very similar to existing works. Conformal quantile regression [Romano et al., 2019] also uses conditional quantile predictors. The method by Stutz et al. [2022] makes similar approximations to define an optimization procedure.

**Questions For Authors:**

1.	What are the differences between the analysis done in this paper and that of Romano et al. [2019] to develop conformal quantile regression? The paper states "...although CQR gives similar results to our method in some situations [...], it has the drawback to not assess the uncertainty of a particular prediction model $\hat{f}$." (lines 411-414, column 2). Why is this a drawback? Arguably, learning end-to-end adds benefits.

2.	What are the differences between the algorithmic choices in this paper and that of Stutz et al. [2022]?

3.	How is Collarary 3.3 different from other bounds like Lei and Wasserman [2013]?

4.	In Proposition 3.1, what happens when $(n_{c} + 1)(1 - \alpha)$ is an integer? Will the same proof not extend to this setting?

5.	Is it possible to experimentally validate the theoretical results?

6.	Isn't splitting the data across steps 1 and 2 of Ad-EffOrt better to avoid overfitting?

Refereces

J. Lei and L. Wasserman. Distribution-free prediction bands for non-parametric regression. Journal of the Royal Statistical Society Series B: Statistical Methodology, 76(1):71–96, 07 2013.

**Relation To Broader Scientific Literature:**

The analysis and techniques used are very similar to existing works, notably Romano et al. [2019] and Stutz et al. [2022]. Because of this, the contribution of this paper feels limited to Theorem 3.7 only.

**Theoretical Claims:**

1.	I checked the proofs for Proposition 3.1, Corollary 3.3, and Theorem 3.7.
2.	For Theorem 3.7, what does the distribution of $Y$ being atom less imply? Why is that required?
3.	Is there a proof for Proposition 3.6?

---

> ### Author Rebuttal · Authors · 2025-03-31
>
> We sincerely thank the reviewer for their constructive feedback and insightful questions. Below, we address each point in detail. We also appreciate the suggestions regarding our real-world data experiments and the approximation of the indicator function—these will be included in the final version using the extra page.
>
> **Questions:**
>
> - *For Theorem 3.7, what does the distribution of $Y$ being atom less imply? Why is that required? In Proposition 3.1, what happens when $(n_c+1)(1-\alpha)$ is an integer? Will the same proof not extend to this setting?*
>
> We address both questions together, as they are closely related. In both Theorem 3.7 and Proposition 3.1, at least one of the two conditions must hold. This requirement stems from a technical detail in our proofs, where we rely on an inversion property of the quantile function with the cumulative distribution function (e.g., line 583). This inversion does not behave well when the distribution has atoms, requiring careful handling. In the context of regression, assuming that $Y$ is atomless is a reasonable and often natural assumption.
>
> - What are the differences between the analysis done in this paper and that of Romano et al. [2019] to develop conformal quantile regression?
>
> There seems to be a misunderstanding. In Romano et al. [2019], the authors construct prediction intervals by performing quantile regression of $Y$ given $X$ at levels $\alpha/2$ and $1-\alpha/2$, without learning a scalar prediction function. In contrast, our approach first learns a prediction function $f$ by minimizing the empirical $(1-\alpha)$-QAE, and then (for Ad-EffOrt) estimates the quantile function of the residuals $|Y-f(X)|$ given $X$ to construct the prediction interval. While both methods involve quantile regression, their objectives differ. Additionally, Romano et al. [2019] do not theoretically analyze the size of the resulting prediction sets.
>
> - *What are the differences between the algorithmic choices in this paper and that of [Stutz et al. [2022]](https://arxiv.org/pdf/2110.09192)?*
>
>
> Stutz et al. [2022] propose a method to incorporate conformalization directly into the learning phase by minimizing inefficiency through a differentiable loss. Key differences with our work include:
>
> 1. Their focus is on classification, while we consider regression with residual scores.
>
> 2. Our method directly optimizes the $(1-\alpha)$-quantile of the scores, whereas they define and minimize an efficiency loss, necessitating an additional data split. Additionally, we use a different technique for the smoothing of the quantile function.
>
> 3. Their approach requires a more complex algorithm, making theoretical analysis of prediction set sizes more challenging, whereas we provide explicit theoretical results on set sizes.
>
> Overall, while both approaches share conceptual similarities (e.g., smooth quantile computation), the methodologies differ significantly.
>
> - *How is Collarary 3.3 different from other bounds like Lei and Wasserman [2013]?*
>
> Could the reviewer be more precise by providing an equation to which he/she refers to in Lei and Wasserman [2013]? After a carefull check, it seems that this paper contains mostly asymptotic results, holding for their specific KDE-based estimator, but we might have missed something.
>
> - *Is it possible to experimentally validate the theoretical results?*
>
> We made additional experiments to validate the theoretical result of Prop. 3.1. In the two following figures: https://ibb.co/G4MPB244, https://ibb.co/MDJDG33J, we illustrate the bound when $\delta=.2$ and $\alpha=0.5$. The first figure displays the evolution of one realisation of $\lambda(\hat{C})$ (in blue) and the rhs term of Eq. (7) (denoted $2Q$, in orange) with respect to the number of calibration points $n_c$. The second figure displays an histogram of the distribution of $\lambda(\hat{C})$ when $n_c=200$. The red line corresponds to $2Q$, the rhs term of Eq. (7).
>
> - *Isn't splitting the data across steps 1 and 2 of Ad-EffOrt better to avoid overfitting?*
>
> This is a valid point. Splitting the data could improve generalization and reduce the prediction interval size. Theoretically, this could also facilitate deriving a bound similar to Theorem 3.7. However, even if overfitting occurs during training, the calibration step naturally corrects it by increasing the size of the prediction intervals.
>
> - *Is there a proof for Proposition 3.6?*  Yes, the proof is provided in Appendix B.2.
>
> **Other Related Work:** We appreciate the additional references. As our work focuses on regression, we had not included classification-specific papers; however, we will incorporate them in the final version. Most of the other cited works are already referenced in our paper, except for Dhillon et al. [2024], for which we refer the reviewer to our response to Reviewer d77L.

---

### Decision · Program_Chairs · 2025-05-01

**Decision:**

Accept (poster)

**Comment:**

This paper theoretically studies the efficiency of prediction sets/intervals in the context of conformal prediction. In the case of split CP, the paper highlights the importance of the quantile absolute error (QAE) to reduce the set size at calibration phase, leading to two algorithms,
EffOrt and Ad-EffOrt. Reviewers agree that the paper does a nice job of leveraging insights from theoretical results to propose a new algorithm.
One critical concern is that the paper overlooks many existing works. The authors did a nice job in responding to reviewers' comments, leading to increased overall scores.